# Why Should Growth Hormone (GH) Be Considered a Promising Therapeutic Agent for Arteriogenesis? Insights from the GHAS Trial

**DOI:** 10.3390/cells9040807

**Published:** 2020-03-27

**Authors:** Diego Caicedo, Pablo Devesa, Clara V. Alvarez, Jesús Devesa

**Affiliations:** 1Department of Angiology and Vascular Surgery, Complejo Hospitalario Universitario de Santiago de Compostela, 15706 Santiago de Compostela, Spain; 2Research and Development, The Medical Center Foltra, 15886 Teo, Spain; pdevesap@foltra.org; 3. Neoplasia and Endocrine Differentiation Research Group. Center for Research in Molecular Medicine and Chronic Diseases (CIMUS). University of Santiago de Compostela, 15782. Santiago de Compostela, Spain; clara.alvarez@usc.es; 4Scientific Direction, The Medical Center Foltra, 15886 Teo, Spain

**Keywords:** GH and eNOS, IGF-I, oxidative stress and arterial inflammation, vascular homeostasis, neovascularization, arteriogenesis, GHAS trial

## Abstract

Despite the important role that the growth hormone (GH)/IGF-I axis plays in vascular homeostasis, these kind of growth factors barely appear in articles addressing the neovascularization process. Currently, the vascular endothelium is considered as an authentic gland of internal secretion due to the wide variety of released factors and functions with local effects, including the paracrine/autocrine production of GH or IGF-I, for which the endothelium has specific receptors. In this comprehensive review, the evidence involving these proangiogenic hormones in arteriogenesis dealing with the arterial occlusion and making of them a potential therapy is described. All the elements that trigger the local and systemic production of GH/IGF-I, as well as their possible roles both in physiological and pathological conditions are analyzed. All of the evidence is combined with important data from the GHAS trial, in which GH or a placebo were administrated to patients suffering from critical limb ischemia with no option for revascularization. We postulate that GH, alone or in combination, should be considered as a promising therapeutic agent for helping in the approach of ischemic disease.

## 1. Introduction

The fact that growth hormone (GH) is a necessary actor for many physiological processes and a real breakthrough for the treatment of many pathological situations beyond simple human longitudinal growth does not need justification [1]. Today, GH is considered a key hormone that acts in virtually all organs and tissues, in which it performs important specific functions. From the current knowledge of GH actions, it can be inferred that, overall, this hormone is a hormone for cell proliferation and survival. The persistent GH secretion out of the growth period is clear proof of the importance of the actions of this hormone at multiple levels, such as the gonads, liver, kidneys, nervous system, adipose tissue, skeletal muscle, and bone, as well as on the cardiovascular, hematopoietic, and immune systems [1].

In fact, virtually all organs and tissues have receptors for this endocrine hormone, and the hormone is also produced in practically all the cells of the organism, where it plays specific autocrine/paracrine roles [1]. However, the sometimes-contradictory results of the use of GH in clinical trials only reflect how little we know about how this hormone really works and how dependent it is on the physiological or pathological state and the microenvironment in which it is acting or the dose or time during which GH is administered. In this review, we analyze the effects of GH on the cardiovascular system, particularly on vascular homeostasis. Since IGF-I is an important mediator of GH actions, the role of IGF-I on vascular homeostasis is also analyzed.

Much evidence supports the participation of local or circulating GH in vascular homeostasis, because when a deficiency of this hormone is present, an endothelial dysfunction appears with serious consequences; this is most likely the reason by which untreated GH-deficiency (GHD) is associated with an increased risk of atherosclerosis and vascular mortality, while GH treatment may reverse early atherosclerosis [2,3,4,5]. A recent study in subjects without GHD or any cardiovascular disease (CVD) but with one or more CV risk factors (age, smoking, obesity, hypertension, dyslipidemia, and insulin resistance) demonstrated that GH and its mediator IGF-I play a protective role in arterial wall changes associated with vascular aging [6]. In fact, receptors for GH (GHR) and IGF-I (IGF-IR) are expressed in the vascular endothelium [7,8,9], and some studies have suggested that GH itself is expressed in this special gland of internal secretion [10,11]. These data indicate that GH and IGF-I have to play a very important role in the maintenance of normal endothelial function. Endothelial dysfunction in GHD has been demonstrated by an impaired flow-mediated dilation, which improved with GH treatment [12], indicating that GH played a role in vascular reactivity [13], as had been shown by the group of Napoli [14]. Additionally, GH treatment in GHD patients has also been found to lead to the normalization of high arterial wall thickness and arterial stiffness in these patients [5], and it normalizes a series of markers of endothelial dysfunction that are generally increased in untreated GHD patients [15]. 

This brief introduction allows us to understand the very important role that GH plays in the cardiovascular system, as it has been reviewed in several occasions [9,16,17]. Detrimental changes in aging arteries negatively influence the ability to compensate after arterial occlusion [18], something that has to be highlighted as occurring parallelly with the GH decline that is experienced as we get older [1,19,20]. As is known, redox imbalance during aging is responsible for this negative effect on arteries, and GH exerts many positive effects for counteracting it, as is demonstrated throughout the text.

Hence, we review here how GH and its mediator IGF-I can act in the arterial wall to favor normal physiological functioning, as well as how both molecules play an important role in collateral remodeling after arterial occlusion. In addition, we bring to light some surprising data that may have been overlooked so far in arteriogenesis with the aim of improving the understanding of, not only of the typical role attributed to GH in the induction of endothelial nitric oxide synthase (eNOS) and the production of NO, but also some ideas about the role of the redox system in the control of homeostasis and vascular remodeling and to clarify how the vessels respond to shear stress forces (SSF) to increase their final size with the participation of the GH/IGF-I system. Arteriogenesis will be described underlining those aspects in which GH could help. Finally, we present some molecular data obtained from the GHAS trial about the benefit of using GH as a rescue therapy in real patients with critical limb ischemia without options for conventional revascularization. 

## 2. Vascular Homeostasis: Role of the GH/IGFI Axis 

A normal embryonic development needs the formation of blood vessels [21]; after birth, there is also the need of the formation of new blood vessels while growing, as well as in some physiological processes such as the menstrual cycle in women and the development of the mammary gland during pregnancy [22]; however, apart from these situations, neovascularization rarely occurs in adulthood, and, when it occurs, it is associated with pathological settings such as wounds, muscle injuries, fractures, tumors and hypoxia/ischemia. 

As seems logical, hypoxia is a very important stimulus for the growth of new blood vessels [23], triggering this growth through hypoxia-inducible factors (HIF) that act on the expression of pro-angiogenic factors but also of anti-angiogenic factors to achieve the perfect number and size of the vessels necessary to compensate for the lack of oxygen supply and to regenerate the tissue [23]. This is a clear and well-established fact of angiogenesis. However, when a progressive narrowing of the vascular lumen appears, preexisting collaterals have to grow to compensate for the lack of distal flow, which is triggered in a totally different way as described below. In any case, the stimulation of eNOS that leads to the production of nitric oxide (NO) from the vascular endothelium seems to be the key mediator for both kind of reparative processes. NO is a potent vasodilator, and it therefore increases blood supply to the zone affected by hypoxia/ischemia. This implies changes in the vascular tone, vasorelaxation, and vasopermeability, which are affected in arteries that suffer for atherosclerotic damage. Interestingly, GH activates the NO pathway [14,24] through direct mechanisms that seem to be specific and independent of the GH-mediator IGF-I [24], although initial studies had indicated that the elevated plasma levels of IGF-I increase NO release in cultured endothelial cells (ECs) [25,26], and more recent data have shown that both GH and IGF-I regulate the expression of eNOS in the aorta of hypophysectomized rats [27]. Moreover, IGF-I has been shown to decrease the release of pro-inflammatory cytokines, such as IL-1β and induce the release of anti-inflammatory cytokines such as IL-10, at least in a model of induced pancreatitis [28]. At this point, it seems to be of interest to indicate that the GH-secretagogue ghrelin also induces vasorelaxation by stimulating eNOS expression in GHD rats [29]. Many studies have clearly shown that members of what we might call the GH system (GH itself, IGF-I, GHR, IGF-IR, GH-secretagogues, and inhibitors of GH-signaling pathways) play a key role in vascular homeostasis.

The mechanism by which GH acts at this level is explained further in the text and is schematized in Figure 1. However, it is necessary to highlight that endocrine GH interacts with its membrane receptor GHR and activates the associated JAK2/STAT5 (Janus kinase 2/signal transducer and activator of transcription 5) and Src family kinases, leading to a cascade of tyrosine phosphorylation responsible for mediating most of the effects of GH at the genomic level; it is especially involved in GH-induced cell proliferation (for a more detailed explanation see [30,31]) and has been postulated as responsible of eNOS activation [32], although this has not been demonstrated. Another key signaling pathway activated by tyrosine phosphorylation after the interaction GH–GHR is that of PI3K/Akt (phosphoinositide 3-kinase/serine-threonine kinase), which is stimulated after activation by JAK2 of the insulin receptor substrate (IRS) and is an inducer of eNOS activation and NO production [32]. Activated extracellular signal-regulated kinase (ERK) translocates into the nucleus and regulates the expression of genes involved in cell proliferation, differentiation, and survival, but it also, and perhaps more significantly, regulates cell motility and migration [33], a mechanism that is very important for the formation of new vessels, as we describe later. The interaction GH–GHR also induces the activation of the focal adhesion kinase (FAK), which responsible for the reorganization of the cytoskeleton in many cell types [34,35], although its effects on vascular ECs have not yet been demonstrated.

Given the effects of GH on the production of endothelial NO, this GH-induced NO could be expected to produce a toxic effect on the vascular endothelium in situations of high levels of oxidative stress, since in these situations, NO can be transformed in superoxide ion (O_2_^–^) that leads to the production of peroxynitrite (ONOO^–^) and the hydroxyl radical OH(^*^), which have toxic effects in the endothelium [36,37]. Both molecules are encompassed into the term hROS or highly reactive oxygen species. As we see later in the text, the exact mechanism of the increase of NO bioavailability by GH, although not fully understood, could be mediated by modifying the action of prooxidative enzymes such as NOX4. This is in agreement with previous studies that have demonstrated that GH has a protective effect on mitochondria [38,39], which are the main source of oxidants within cells. This is the reason why mitochondrial dysfunction leads to a greater generation of hROS, which, in turn, contributes to the presentation of a senescent phenotype in ECs and to the activation of redox-sensitive transcription factor nuclear factor-kappa B (NF-κB) [40], which decreases endothelium-induced vasodilation and increases the expression of inflammatory genes in the vasculature of aged rats and elderly humans [41,42]. However, as indicated above, plasma physiological levels of GH and IGF-I modify the intracellular levels of oxidative stress [13,43,44]. These concepts are schematized in Figure 2. 

In vitro studies have shown that GH and GHR are capable of translocating to mitochondria through a pathway constituted by caveolae [45,46]. In isolated mitochondria, high concentrations of GH can decrease mitochondrial O_2_^-^ production, most likely inducing a decrease in the activity of complexes II and IV of the mitochondrial respiratory chain [46] and modulating the mitochondrial membrane potential critical to maintain the physiological function of the respiratory chain to generate ATP, because when this membrane potential collapses, the cytochrome C is released in the cytosol, and, therefore, the mitochondrial respiratory chain is affected (Figure 2). In fact, in mice with decreased GH production, such as Ames dwarf mice, the production of hROS by mitochondria is increased in aortas and NO production is decreased in comparison to the oxidative stress observed in wild type animals with normal GH secretion; moreover, treatment with GH in cultured wild-type mice aortas and human coronary arterial endothelial cells has been found to significantly reduce mitochondrial hROS and to upregulate SOD (superoxide dismutase) genes and eNOS, providing an antioxidant phenotypic change [47]. This agrees with previous data showing that GH significantly reduces the intracellular production of hROS and increases NO release in cultures of a human endothelial cell line [8].

Therefore, it is clear that GH plays a positive role in the homeostasis of the vascular endothelium, not only by increasing NO production but also by protecting the endothelium, acting at the mitochondrial level, modulating oxidative stress. However, it has been described that this effect of GH does not occur acutely—rather, it occurs after a time, at least in studies performed in vitro [7,48]. Though this effect has not been properly studied in humans yet, we now provide some important insights from the GHAS trial that support this action. 

Since, as indicated above, GH is expressed in ECs, where it plays an autocrine/paracrine role, one might think that the loss of the endocrine secretion of GH as we age could be compensated for by the cellular expression of this hormone, but this has been only seen in breast carcinomas [11], in which endothelial GH stimulates the proliferation, migration, survival, and capillary formation of ECs. Hence, there are no data to prove the possibility of an effect of autocrine/paracrine endothelial GH substituting the effects of endocrine GH in normal subjects. In addition, it is unknown how GH expression is regulated in vascular ECs and whether the decrease in the endocrine production of this hormone, in old people for instance, is accompanied by a deficit in its cellular production.

Another important effect of GH to explain its action on the vascular system is that exerted on mesenchymal stem cells (MSCs). These cells have been thought to proceed from the bone marrow, where they were identified in the past century [49], but we currently know that they can be found in virtually all adult tissues [50,51,52], particularly in adipose tissue, where they seem to be located in a perivascular zone close to pericytes and ECs [53,54]. This is very important because the external layer of blood vessels, also called tunica adventitia and being for long relegated as a mere support of the tunica media, is currently known to participate in vascular remodeling, since its cells can be activated in response to injuries [55,56,57], hypoxia [58], and hypertension [59]. Though the activation of adventitial progenitors mainly results in proliferation and differentiation into myofibroblasts that migrate into the inner layers of the vascular wall, they also release paracrine factors that regulate vascular remodeling [60]. MSCs also can differentiate into ECs in the presence of the vascular endothelial growth factor A (VEGF-A) [61,62], a factor that plays many roles in cell differentiation, proliferation, and angiogenesis. The relationships between GH and VEGF-A are not well known, but GH plays a pivotal role among the factors that regulate VEGF family expression in humans [63]. In addition, GH regulates the expression of different genes involved in Notch-1 signaling, at least at the ovarian level [63], and two Notch-1 ligands—delta-like protein Dl14 and Jagged 1—that directly regulate angiogenesis in the endothelium [64,65]. Therefore, given the relationships existing between Notch-1, VEGF-A, Notch-1, and GH, it is presumable that the differentiation of MSCs into ECs for neovascularization and endothelium repair may involve the participation of GH (for a more detailed explanation, see Figure 9 in reference [61]).

For vascular homeostasis to take place correctly, a negative regulation mechanism is also needed to prevent an excess of intracellular signaling by GH after its interaction with its membrane receptor. The negative regulation of GH signaling is mainly carried out by the intracellular protein tyrosine phosphatases 1B and H1, suppressors of cytokine signaling (SOCS, 1, 2, and 3), sirtuin 1, activated STAT protein inhibitors (1, 3, and 4), cytokine-induced SH2-containing protein (CIS), and tyrosine-protein phosphatase non-receptor type 6 (PTPN6 also called SHP-1) [66] (Figure 1). Given the different systems that act on the negative regulation of GH signaling pathways, there must be a perfect balance between them to prevent a pathological situation from occurring.

In summary, as we have just seen, GH plays a very important role in vascular homeostasis, acting on NO production, protecting from oxidative stress, regulating cell proliferation, differentiation and survival, regulating cell motility and migration, and inducing the differentiation of MSCs into vascular ECs. Since GH secretion decreases until it almost disappears as we age, it seems logical to assume that the loss of the hormone is one of the main causes of vascular problems that occur in old age.

## 3. Molecular Roles of GH/IGF-I in the Vascular Wall that Favor Arteriogenesis 

### 3.1. GH/IGF-I Response to Shear Stress Forces (SSF): the Mechanosensing Pathway

In 2011, W. Schaper highlighted two key aspects of the arteriogenesis pathways: 1) only the inhibition of the total production of NO could inhibit collateral growth; 2) the mitogenic agent for the activation of smooth muscle cells (SMCs) from the vascular wall was still unknown. Though several candidates were given, mainly fibroblastic growth factor (FGF) and platelet-derived growth factor (PDGF), none met all the characteristics required [67]. However, considering the key role of the GH/IGF-I axis in the vascular homeostasis for regulating eNOS and NO, as stated above, and since both GH and IGF-I are potent mitogens [68] that are even stimulated by PDFG, perhaps both hormones should be considered candidates to be that unknown mitogenic agent or agents, because there might be more than one. Throughout this review, we try to present the evidence for this statement.

Since the discovery of the SSF as the main stimulator of collateral enlargement during arteriogenesis, many investigations trying to find the connection between mechanical and biological aspects have been developed. The genes involved in shear stress, activated by the stimulation of mechanical endothelial receptors, have been proposed as sensitive elements to increase the redirected flow through collaterals after arterial occlusion. However, some aspects have not yet been elucidated, as is the case of which molecules mediate the mechanical signaling pathway; in other words, are there factors involved in the arterial growth capable of being sensitive to mechanical forces? The answer to this question should be yes, and some of these factors could be locally produced hormones. Evidence accumulated in the last years has shown how the GH/IGF-I system can be regulated by multiple factors, such as growth factors, cytokines, lipoproteins, reactive oxygen species, hormones, neurotransmitters, and hemodynamic forces [69]. As is known, many patients with arterial hypertension develop left ventricle hypertrophy and aortic wall thickening, while patients suffering an aorto-cava fistula develop the hypertrophy of the right ventricle and an overload of the vena cava. Intuitively, one has to realize that something must orchestrate this adaptation to the pressure or the changes produced by volume. Given the strong mitogenic capacity of GH and IGF-I and the fact that there are many receptors for these hormones in large vessels, such as the aorta or vena cava [68,70,71], there should be a cross-talk between these hormones and hemodynamic forces. This hypothesis was demonstrated in an interesting study in which a volume or pressure overload was applied to rats to study the gene expression of GH receptors (GHR) and IGF-I mRNA in the vena cava and aorta. In the distended volume model of vena cava, 8-fold and 3.5-fold increases in IGF-I and GHR mRNA levels, respectively, were found compared to control animals, as early as day four. Additionally, the IGF-I protein was located in SMCs. In the aortic stress model, 4-fold and 5-fold increases in IGF-I and GHR mRNA, respectively, were found in aortas under pressure on day seven. Both vena cava and aorta showed structural adaptations with a growth response. This study was very important, since it represents a possible way of connection between the mechanical and biological pathways, with the autocrine/paracrine production of GH/IGF-I playing a major role. The increase in vascular wall stress, therefore, seems to trigger the overexpression of IGF-I and GHR mRNA in large vessels [71]. These data are consistent with the finding in a model of aortic coarctation in rats, in which an increase of more than double IGF-I levels was detected in blood vessels on day seven that was persistent on day 21 and was accompanied by a growth of SMCs and ECs [72]. Thus, in this model, IGF-I plays a pivotal role in the wall vessel remodeling of the aorta under high shear stress. Even more, the authors stated that IGF-I mRNA levels were also elevated in quiescent aortic SMCs, which underlines the role of IGF-I as an autocrine growth factor for dynamic changes in the vascular wall [73]. An earlier report also showed the same data but in a rat model of femoral artery overload, where a strong immunoreactivity for IGF-I was detected in the middle layer of the left femoral artery 24 hours after the right femoral ligation, along with a significant decrease in the expression of IGF-I in the occluded artery in the zone distal to the occlusion [74]. All these findings provide evidence for a major role of autocrine/paracrine GH/IGF-I system in mediating vessel wall growth during shear stress (Figure 3). 

To perform its mitogenic function, GH has been shown to directly stimulate Src family kinases (SFK), which in turn activate mitogen-activated protein kinase 1 (also known as extracellular signal-regulated kinase 2 (ERK2) and mitogen-activated protein kinase 3 (also known as ERK1) through a phospholipase Cγ–Ras pathway [75]. The prolactin (PRL) receptor similarly activates the same SFK signals [68], and it is well known that GH interacts with the PRL receptor with the same affinity that it does with its GHR [76].

All these adaptations in the vascular wall by GH/IGF-I may be supported by the fact that GH is capable of modifying the aortic content and composition of collagen and elastin, and even its mechanical behavior [77]. A decrease in collagen and elastin in the aorta wall can be detected in aged rats and humans in parallel to the decrease in GH/IGF-I levels, although plasma IGF-I is still maintained at lower levels, since its liver production does not exclusively depend on GH, which virtually disappears in the elderly [1]. In fact, the administration of GH to aged female rats was found to augment the collagen deposit by 300% and its turn-over in the media layer. It seems that the main mediator of this GH effect in the aorta is IGF-I, produced locally in SMCs. In the thoracic aorta of aged female rats, GH was found to increase the ratio of collagen I/III, improving both the stiffness in those areas under high overload and the extensibility in those with low overload, thus adjusting the aortic mechanical characteristics to the overload [77]. These studies shed some light on the idea that local hormone production participates in the regulation of structural adaptations of blood vessels under stress conditions; for this, the IGF-IR number plays a major role for SMCs response. On the one hand, the increase in the expression of IGF-IR by GH or other factors such as angiotensin II or FGF-2 is crucial for the mitogenic effects of IGF-I on SMCs. On the other hand, factors such as Tumor Necrosis Factor alpha (TNF-α) or ox-LDLc decrease IGF-IR, favoring SMCs apoptosis. Effectively, GH and IGF-I play a key role in vessels for the growth and survival of SMCs, inhibiting apoptosis produced by mitochondrial dysfunction induced by TNF-α and ox-LDLc [69].

### 3.2. GH and GHR: a Complex Regulation that Explains Different Results in the Studies Performed

Given the fact that many GHRs have been found in the vascular system, this organ must be a special target for GH. In addition, it seems to be clear that many of the effects of GH on the vascular wall are independent of IGF-I [7,8,76,78]. Indeed, although the GH/IGF-I axis is highly coordinated to act in different tissues, both GH and IGF-I may have independent actions [79]. The type of secretion of GH is another factor that influences the different actions of the hormone. That is, while pituitary GH is secreted in pulses, some of them of great amplitude, autocrine GH is produced in small quantities and almost continuously. These differences are responsible for the different effects that both types of GH induce. For example, the oncogenic potential of GH seems to be associated only with local secretion and not with the exogenously administered or pituitary hormone [11]. Effectively, both types of GH production regulate gene expression in different ways [80], a fact that also implies differences in the GHR-related signal transduction pathways [81]. 

Though GHRs have been found in different arterial layers, the endothelium seems to be the place where these receptors have been found in a greater concentration. Interestingly, the GHR gene is a hypoxia-inducible gene, since the overexpression of HIF-1α (hypoxia-inducible-factor-1-alpha) increases its expression, suggesting that GHR may be involved in hypoxia-induced processes, such as the generation of new vessels [82], although many factors can modulate the expression of the GHR gene [83]. However, many of the GH actions in the artery media layer are mediated by IGF-I, especially in SMCs [77], or they occur as a consequence of the diffusion of GH-generated signals from the endothelial layer, such as the diffusion of NO. 

One of the most interesting locations of GHR that supports the role of the hormone in physiological arteriogenesis during adulthood is in the vessels of the endometrium, where GH seems to play an important role favoring the creation and maintenance of vessels during every sexual cycle in fertile women [63,84]. 

As widely described, GH induces eNOS expression and NO release in cultured human ECs [8]. When L-nitroarginine methyl ester (L-NAME) is administered, GH loses its action on NO, which confirms this main action on the endothelium [85]. To produce NO, eNOS has to be phosphorylated on serine 1177 via PI3K/Akt and activated by a calcium-calmodulin [86]. Diffusing NO triggers a soluble guanylyl cyclase into SMCs, increasing intracellular levels of cyclic GMP (cGMP) that activates protein kinase G1 via the phosphorylation of the inositol-triphosphate receptor-associated cGMP kinase and the sarcoplasmic reticulum ATPase. The consequence is that intracellular calcium decreases leading to vascular relaxation and modification of the arterial tone [86,87], one of the most important action of NO and GH, demonstrated after the arterial infusion of the hormone [14]. It has to be underlined that eNOS activity is highly regulated in cells at the transcriptional level and by other factors such as acylation and phosphorylation, or by protein–protein interactions [86]. However, vasodilation is not the only benefit of NO on the vascular wall, since it facilitates vasopermeability (by decreasing VE-cadherin activity) and the chemoattraction of monocytes, and it reduces lipoxygenase activity, ox-LDLc, platelet adhesion, and SMC proliferation and migration [13,88]. As described above, NO is the main route used by GH to contribute to vascular homeostasis. 

A high concentration of GHRs has been detected in the aorta, femoral, and carotid arteries, where it mediates many GH effects by activating the JAK/STAT pathway, among others [68,70]. GH cannot induce the transcription of IGF-1 in ECs [76], but this does not mean that GH does not interact with IGF-I in the artery media layer as stated, where it precisely regulates the growth and survival of SMCs mediated by IGF-I, for which SMCs have many receptors [69]. GHR is crucial for the wide effects of GH, allowing the hormone to internalize in cells where GH triggers many signaling pathways and influencing gene transcription. This internalization is facilitated by GH-binding proteins (GHBP), the extracellular component of GHR. The concentration of GHRs determines the intensity of GH signals, mainly when considering the eNOS–NO pathway. The higher the number of GHRs, the greater the effect of GH. Nevertheless, there is a limit in which if more hormone is administered the effect not only does not increase but decreases. For instance, it has been shown that 1 nmol/L of GH produces 6.39 µmol/L NO, 10 nmol/L of GH produces 6.45 µmol/L NO, but 100 nmol/L of GH produces 6.38 µmol/L NO [89,90]. It is important to point out that GH itself controls the gene involved in the expression of its receptor and that this regulation depends on the time and dose of hormone administration. This fact is important to understand the reasons why GH has different effects depending on the dose and to understand why clinical trials with GH sometimes differ in their results. As explained before, the relationship between GHR and GH is bidirectional, and GHBP binds to the free hormone in the blood to control the amount of free GH that interacts with its receptor but also to regulate GH clearance. Even inside the cells, GH actions are exhaustively regulated. This is an important concept common to all the growth factors acting in the organism. The effects of GH on GHR gene expression seem to be dependent on the time/dose of exposure in addition to the cell type and whether the experiment is carried out in vivo or in vitro. However, GHR gene expression is even more complex, as it is also influenced by other factors such as nutritional intake, steroids, or diabetes mellitus. Furthermore, the GHR gene has multiple 5’ untranslated exons that are controlled by multiple promoters, thus showing the complexity and high quality regulation of the gene expression of this receptor [83].

### 3.3. GH/IGF-I may Favor Inflammation during Collateral Enlargement

During collateral enlargement, two types of cells are activated to achieve the great change, and both endothelial and SMCs acquire a proliferative and secretory phenotype after a previous phase of NO-dependent vasodilation because hemodynamic or short-term compensation always runs before than that originated by growth factors or long-term compensation. While flow and metabolism control is mainly carried out at the local level in physiological situations, when a pathological condition such as ischemia is present, a systemic response (centered on redistribution of flow and microvascular adaptations) is usually required [91]. That needs the neurohormonal axis effects. In both circumstances, local and systemic GH/IGF-I is important for regulating short and long-term adaptations. 

The inflammation of the collateral wall aims to achieve vascular remodeling. To do this, two stages can be distinguished: First, ECs facilitate the inflammatory response in the vascular wall; second, SMCs proliferate to increase the vessel size. It is not the aim of this chapter to describe the whole process of arteriogenesis. However, it is necessary to highlight those aspects in which GH could participate. For example, it has been described how monocytes, monocyte chemoattractant protein-1 (MCP-1), and lymphocytes are crucial for vascular remodeling during arteriogenesis in the inflammatory phase [92], since this process is not complete or is delayed when all these factors are not present. It has been shown that GH strongly induces these cells and proteins [1,93,94], playing a pivotal role in the chemotaxis and migration of human monocytes. 

MCP-1 has been found elevated in blood samples from the collateral network of coronary arteries [95], and it has been found to be transiently and selectively increased in the ischemic muscle during the first three days after ischemia [96], which underlines its importance in arteriogenesis. The relationship between GH and MCP-1 has been conveniently studied and seems to be mediated by the activation of both JAK2 and p44/42 mitogen-activated protein kinase (MAPK), since MCP-1 significantly decreases in cells pretreated with the JAK2 inhibitor AG490 or MAPK inhibitor PD98050 [94]. In the referred experimental study, MCP-1 mRNA levels increased up to 8 times after the administration of a low dose of GH [94]. As is known, both MCP-1 and cell adhesion molecules (CAMs) play central roles in igniting the arteriogenic process. The former play roles in the attraction of inflammatory cells, and the latter play roles in the adhesion and invasion of the vascular wall by this type of cells. 

CAMs are also related to GH, especially vascular cell adhesion molecule-1 (VCAM-1). GH significantly increases the expression of VCAM, as demonstrated when serum from healthy patients treated with the hormone is administered to cultured umbilical vein ECs [97]. An indirect mechanism has been proposed to explain this action, one perhaps mediated by VEGF, IGF-I, or SDF-1 [16,76,91,98], factors that upregulate CAMs [92]. This action has also been observed in adults with GHD, in which GH replacement therapy significantly increases VCAM-1. Therefore, both MCP-1 and VCAM could be modulated by GH, facilitating the first phase of arteriogenesis. In addition, CD34+ cells, also involved in arteriogenesis [99], are stimulated by GH, both in number and function. In fact, circulating CD34+ are decreased in adults with GHD, which contributes to the endothelial dysfunction detected in these patients. GH replacement therapy improves this dysfunction by correcting CD34+ cell depletion [100].

Almost all human immune cells express GHRs and produce its ligand that may act in an autocrine or paracrine manner [101,102]. GH produced by immune cells has the same structure as that GH found in the anterior pituitary gland [103]. In addition, immune cells have IGF-IRs and produce IGF-I. This also supports the fact that GH could mediate the arteriogenic process, since monocytes and T-lymphocytes are key cells for the inflammatory response needed for collateral enlargement [104,105] (Figure 3 and Figure 4). GH can favor the migration and invasion of immune cells to the vascular wall, and once in the vascular wall, GH might directly or indirectly activate them to produce cytokines.

It is noteworthy that monocytes have to be activated to help in arteriogenesis, since when they are transplanted directly from the blood in animal models of ischemia, they do not influence this process. However, when these cells are previously activated by the monocyte colony stimulating factor (M-CSF), collateral growth begins [108]. Thus, M-CSF favors a suitable environment for stable monocytic function [18]. It has to be highlighted that GH is strongly related to M-CSF and that in vitro and in vivo studies have shown that GH activates monocytes to produce cytokines and stimulates the chemoattraction and random migration of the same even at picomolar concentrations of the hormone [109]. The cellular pathway involved was demonstrated many years ago [110], as GH stimulates the tyrosine phosphorylation of two proteins, p130Cas and CrkII, their association and the association of other multiple tyrosine-phosphorylated proteins that end up activating the c-Jun N-terminal kinase/stress-activated protein kinase (JNK/SAPK) [111]. Such a large multiprotein signaling complex is not only the basis for activating monocytes but is also essential for other effects such as cytoskeletal reorganization, cell migration, chemotaxis, mitogenesis, and/or the prevention of apoptosis and gene transcription [110].

Monocytes can also have trophic effects on homeostasis, e.g., secreting CSF during growth [112], suggesting that this factor interacts with the GH/IGF-I axis. Macrophages also produce IGF-I in response to CSF-1 or GM-CSF, and IGF-IRs are highly expressed in macrophages stimulated by the M-CSF family [105,113,114]. Despite this, the specific link between M-CSF family and GH has not yet been properly studied, although it has been seen that many CSF-1-dependent macrophages are present in the pituitary gland in the course of somatotropic cell development [115]. 

### 3.4. GH/IGF-I and eNOS in Arteriogenesis. Insights from the GHAS Trial: is Redox Balance the Main Actor?

As described above, several signaling pathways work together to grow the collateral arteries, and one of the most relevant, although with some contradictory results, is the NO pathway. This pathway has been considered crucial for many years for collateral enlargement, both during the early and late stages of arteriogenesis. Measuring several variables such as perfusion of the limb, the diameter of the collateral artery, and the number and location of pericytes within the ischemic hindlimb, less recovery of blood flow and a smaller collateral artery diameter have been demonstrated in the group of eNOS^–/–^ and L-NAME mice than in the wild-type. The pericytes barely appeared, and they did so in a random pattern in the former groups compared to the wild-type [116]. Another fact that supports the role of this pathway in arteriogenesis is the finding of a lower eNOS expression both in the thigh and in the gastrocnemius muscles in diabetic mice, which leads to reduced collateral enlargement. The reduced expression of eNOS in diabetic mice (types 1 or 2) may contribute to the deficient arteriogenesis and angiogenesis seen in these animals, while the treatment with thiazolidinediones, that increases eNOS activity through the peroxisome proliferator-activated receptor gamma (PPARg) [117], restores these deficient responses [116]. Animal models of ischemia after training support the important role of eNOS and NO in arteriogenesis, with a greater effect than that induced by VEGF, because the latter is a secondary factor for collateral growth and eNOS a primary factor. Conversely, the role of eNOS in angiogenesis is less relevant, but VEGF also needs it to carry out its actions [118]. Therefore, the difference in NO dependence is the key factor that distinguishes between both processes.

However, not all authors agree completely, since it seems that eNOS and NO could be of great importance for the NO-mediated vasodilation of peripheral collateral vessels after arterial occlusion, but their relevance seems to be minor for collateral enlargement. Tissue perfusion and collateral-dependent blood flow was found significantly increased in mice overexpressing eNOS compared to wild-type mice, but only immediately after ligation. In eNOS^–/–^ mice, collateral-dependent blood flow was found to remain poor until day seven, after which it recovered, thus suggesting a delay but not a complete impairment of collateral growth. Besides, no differences in collateral arteries between the three groups of mice were histologically confirmed at the end of the study, and the administration of a NO donor induced vasodilation in the collateral arteries of eNOS^–/–^ mice, but it does not so in wild-type mice [119]. Interestingly, eNOS deficiency may be compensated for by iNOS activity [67], and this might be the explanation of this finding and other different results. That is, the exact role of eNOS is not fully understood. Nevertheless, some insights from the GHAS trial (Eudract 2012-002228-34), performed by our group, could be useful to clarify the activity of eNOS in humans. In this randomized, controlled, phase III trial, recombinant human growth hormone (rhGH) (0.4 mg/day for eight weeks) or a placebo was administered to patients with critical limb ischemia without revascularization options. The main characteristics of the patients studied are outlined in Table 1, while Table 2 shows the specific characteristics of the patients once distributed in each treatment group.

Muscle samples from the calf of the ischemic limb were taken at baseline and just after finishing the treatment of the patients. After analyzing mRNA expression of angiogenesis-related genes, the molecular data showed us a surprising finding: In the group treated with GH (group A, green bars), there was an increase in the level of eNOS mRNA compared to baseline, but this increase was significantly lower than that detected in the placebo-treated patients (group B, blue bars). Parallelly, a significant and strong decrease in NOX4 levels was only observed in the GH group (Figure 5); this means that the activity of eNOS depends largely on the redox balance in the ischemic muscle, or, even more, that the arteriogenic growth of the vessels depends largely on the redox imbalance and that GH, by correcting this stress, also reduces the signal that stimulates eNOS.

This is an important concept because it indicates that there is an insensitivity to NO rather than a real depletion of it or eNOS in patients with peripheral ischemia [120], and this finding matches perfectly with the high redox stress and insensitivity to NO found by other authors in humans and animals with peripheral arterial disease [121,122,123]. This is also consistent with the fact that an increased oxidation can be seen in GHD-adults, secondary to an elevated activity of lipoxygenase activity and ox-LDLc (the main source of atherosclerotic lesion of the arteries) as a consequence of NO depletion. After GH replacement therapy, this redox alteration can be significantly reduced, together with the recovery of a normal flow mediated dilation test, compared to pre-GH administration and control patients [13]. At this point, it is of interest to remark that senescence, frequent in patients with peripheral arterial disease, is like a GHD state. 

Therefore, the important message here is the relevance that redox balance exerts on collateral enlargement. As has been described, while small or acute redox stress, characterized by an elevated production of reactive oxygen species (ROS)/reactive nitrogen species (RNS), seems to positively influence neovascularization, a chronically high level of oxidation is detrimental for vascular remodeling [124]. In the GHAS trial, patients under placebo treatment maintained a high level of NOX4 and eNOS mRNAs in the calf muscle, trying to compensate for the decrease in NO bioavailability. The administration of GH stopped this vicious circus, lowering the redox imbalance and also increasing the bioavailability of NO without the need of an increase in the production of eNOS. Indeed, the bioavailability of NO depends largely on the production of ROS, since they react with NO to inactivate it, thus lowering its levels [32], which means that the negative regulation of redox stress by GH in patients with critical limb ischemia increases the bioavailability of NO without a significant elevation of eNOS in the ischemic skeletal muscle. This unexpected finding highlights the dynamic role of redox balance in vascular homeostasis. Typically, NOX4 is upregulated in cells by SSF, hypoxemia, or cytokines like TNF-α, and the latter is usually elevated in patients with ischemic process [125], favoring the inflammation and phosphorylation of eNOS, which reduces NO production [126]. 

In the GHAS trial, GH also significantly reduced the circulating levels of the inflammatory marker TNF-α in blood samples, which is consistent with other studies [127,128] and with the fact that GH improves redox balance (Table 3). The deleterious effects of TNF-α in these circumstances have been described, including the elevation of intracellular SOCS [91] and the diminishment of IGF-I and SMCs in the vascular wall [69]. GH, as seen, could correct these negative effects, decreasing TNF-α levels. Though proinflammatory activity has also been seen for GH, this is something that occurs when high levels of the hormone are considered, demonstrating, once again, that the role of GH depends on its physiological or pathological concentrations and on whether the morbid condition is stablished acutely or chronically [129]. However, C reactive protein (CRP) did not significantly change at the end of the treatment compared to the placebo. It has to be noted that this marker has a different meaning than TNF-α and that the levels of CRP were found to be significantly elevated in the GH group at baseline compared to the placebo group, which could have influenced the final results. 

Regarding ischemic disease, authors sometime do not agree because they do not describe the same phenomenon, since an acute injury is normally produced in an animal model of ischemia, while a chronic process is usually developed in a human being with peripheral or coronary artery disease. In fact, in the human heart, after acute coronary occlusion, the arteriogenic phenomenon is very fast, just like in animal models, taking between one and two weeks for the great majority of patients [130]; meanwhile, in chronic ischemia, the collateral enlargement takes several months [131]. This is a concept that needs to be highlighted, as in the chronic ischemic state, due to the lower and slower activation of the physiological compensation that is impaired by age, GH could be useful by enhancing all these mechanisms and improving distal flow. 

In a study in which animal models of acute and chronic ischemia were used simultaneously, it was shown that the mechanisms that regulate blood flow recovery, gene expression, macrophage infiltration, and hemangiocyte recruitment are critically different, since they depend on the arterial occlusion rate and the mechanisms that regulate the recovery of blood flow [132]. The most important finding in that study was that MCP-1 or shear stress-induced genes like eNOS or Egr-1 are not sufficiently activated to induce collateral artery enlargement in the model of gradual ischemia, since SSF through collaterals is weaker compared to that in an acute ischemia model; additionally eNOS or MCP-1 are less stimulated in gradual or chronic ischemia than in the acute process at the thigh, the place where collateral growth mainly takes place. Though the upregulation of VEGF and PlGF is found in acute models, particularly important are eNOS and KDR/Flk-1 in the remodeling of vessels [118], while in humans, as stated above, it seems that lowering the redox imbalance could be more profitable. Interestingly, in the GHAS trial, we also found a parallel and significant increase of VEGFA-R2 or KDR/flk-1 mRNA levels from muscle samples in the GH group (group A) compared to the placebo group (group B) (Figure 6A), which confirmed the action of VEGFA, a finding consistent to that from animal models of hindlimb ischemia treated with the IGF-I plasmid [133]. This effect could be dependent of IGF-I, but we hypothesize that it is a direct action of GH on the muscle, because a parallel increase in IGF-I mRNA level has not been found, at least at the time in which muscle samples were obtained (Figure 6B). However, the elevation in KDR/flk-1 was not accompanied by a parallel significant increase of VEGFA mRNA levels after two months of treatment (Figure 6C). This fact could have an explanation. Knowing that VEGFA mRNA levels decrease after stimulation with very low levels within the fourth week [134,135], it is easy to understand that, muscle sample were obtained when the levels of this factor had already declined (at eight weeks), overlooking any possible elevation produced during the previous month. 

This finding is also very important, as it has been advocated that the differences between VEGF action in different tissues might be determined by the number of its receptors rather than by the levels of VEGF [136]. VEGF receptors are usually normal in patients with peripheral arterial disease in the calf [137], albeit with controversial results; meanwhile in acute ischemia, VEGF and VEGFR-2 are diffusely expressed in the affected muscle, as in skeletal muscle that recovers from chronic ischemia, the former factors are restricted to atrophic and regenerating muscle areas [138]. This is consistent with the fact that KDR/Flk-1 expression appears in macrophages and fibroblast cells found in the necrotic area after myocardial infarction [139]. Here, we present clear evidence that VEGF is not normally augmented in distal limb muscles under chronic ischemia, as compared to control muscle samples obtained from limb amputations in non-ischemic patients, and that GH can increase VEGF actions through, at least, a raise of the KDR receptor, also favoring muscle regeneration accompanying vascular development. As is known, KDR mediates most VEGF actions on mitogenesis, survival, and permeability at the vascular wall, being mainly expressed in ECs but also in macrophages and SMCs. This receptor has also been found in myocytes from skeletal muscles, both in the membrane and in the cytoplasm [138], and KDR can be derived from myocytes; from this point of view, GH could act more at the muscular level rather than at the vascular one. However, recent experimental studies could support the fact that GH acts at the vascular level when the KDR receptor increases. First, this receptor appears to be a critical mediator for the action of extracellular RNA (eRNA) and the von Willebrand factor (VWF) released by ECs for the recruitment of leukocytes that initiate arteriogenesis in response to fluid shear stress [140]. Secondly, this receptor also seems to be important for the action of mast cells that, after activation, release VEGFA [141]. Therefore, the finding in our study of an increase in mRNA levels of VEGFA-R2 or KDR by GH must necessarily be related to the facilitation of the action of these molecules. 

In any case, as is known, the increase in SSF after an arterial occlusion triggers the NO pathway, which inhibits the expression of VE-cadherin, which responsible for maintaining vascular membrane integrity and, therefore, increasing vascular permeability and the invasion by inflammatory cells of the vascular wall [142]. Both the disruption of endothelial junctions and the remodeling of the cytoskeleton are necessary factors for vascular permeability and the NO released by the endothelium is crucial. One approximation of the involved molecular mechanism has been shown in a study in which the lack of eNOS reduces VEGF-induced permeability that is mediated by an increase of the Rac GTPase activation. NO depletion impairs the recruitment of the guanine-nucleotide-exchange factor (GEF) TIAM1 to adherent junctions and VE-cadherin, and it reduces Rho activation. NO is crucial for the regulation of the Rho GTPase-dependent cytoskeleton architecture and its action leads to reversible changes in vascular permeability. It seems to be clear that when NO is inhibited, flow-induced arteriogenesis is also interrupted [118].

### 3.5. GH and CXCL12 (SDF-1): A Potential Collaboration for Collateral Growth

Another cytokine that plays an important role in arteriogenesis is SDF-1. This cytokine, produced mainly in platelets, can stimulate collateral growth in several ways. The role of SDF-1 was analyzed in a double animal model of ischemia with both types of processes, acute and gradual, and seems to have a less important function than expected since in the gradual model of ischemia in mice, a smaller increase in SDF-1 was found in both the thigh and calf compared to what occurs in the acute process [132]. However, in another interesting experimental study in mice, an important action of this molecule and its receptor, CXCR4, has been recently been described that could help to understand the real role of this cytokine in arteriogenesis [141]. SDF-1, released by activated platelets, recruits perivascular mast cells by interacting with CXCR-4, and mast cells are responsible for boosting the arteriogenic process by two main mechanisms: On the one hand, they recruit neutrophils, monocytes, and T cells in the vascular wall, and on the other hand, they directly release growth factors, mainly MCP-1 and VEGFA, that stimulate vascular remodeling. Therefore, SDF-1 could promote the inflammatory microenvironment necessary for collateral enlargement by recruiting perivascular mast cells that could play a key role that has never been clearly described. In addition, platelets facilitate the extravasation of neutrophils via GPIbα and uPA receptors, which in turn, activate mast cell degradation by releasing ROS. This is an important finding, and it represents another possible connection between shear stress (mechanical signal) and vascular growth (chemical signal) [141].

From this point of view, particularly striking is the relationship between SDF-1 and GH, since the receptor for SDF-1 (CXCR4) has been found in pituitary somatotrophs, where SDF-1 activates the expression of the GH gene and the production and secretion of GH from the anterior pituitary, regulating the normal physiological function of GH cells. In fact, when SDF-1β was administered to rats, both alone or with GHRH, the production of GH from the pituitary gland was increased by 2.5–3.5 times in a dose-dependent manner [143]. Interestingly, despite the different types of cells in the anterior pituitary gland, only somatotrophs express CXCR4. That means that SDF-1 and GH are closely related, at least at the pituitary level. The cellular mechanisms responsible for this stimulation have been studied in GH4C1 cells, showing two possible pathways: the Ca^++^-independent stimulation of ERK1/2 activity and the Ca^++^-dependent activation of Pyk2 and BK_Ca_ [144]. This positive relation is also supported by the finding that GH stimulates SDF-1, since high levels of the same can be seen in the thymus of GH-transgenic mice and in cultured primary thymic epithelial cells derived from these animals, as compared to age-matched wild-type counterparts. Additionally, thymocytes migration induced by SDF-1 is improved when these cells are exposed to GH [145]. Since both SDF-1 and GH are coordinated during embryological development for vasculogenesis and for immune system function [16,145], it is tempting to speculate that both may act together in situations such as ischemic injuries, mainly stimulating and leading the migration of macrophages, lymphocytes, and, probably, mast cells to the vascular wall during collateral enlargement. 

The finding of Chillo et al. [141] about the possible role of mast cell on arteriogenesis by orchestrating leukocyte function has to be underlined again and could also support our finding of an increase of KDR/Flk-1 in human muscle samples. Mast cells activated by shear stress release VEGFA into the vascular wall that needs its receptor to achieve its action. In addition, as described before, KDR/Flk-1 is a critical factor for mechanotransduction because it induces the release of VWF by ECs. Therefore, GH must be a mediator in this context, enhancing ECs and mast cell activity. 

### 3.6. GH and Mesenchymal Stem Cells (MSCs)

Another intriguing relationship of GH is that which the hormone has with MSCs. These cells migrate in the embryo to the newly formed vessels to release growth factors and stabilize the new vascular network, since they can differentiate into pericytes. However, these cells do not disappear in the postnatal period, and they can be identified in adults in many tissues, especially in the bone marrow but also in adipose tissue or in muscles near vascular structures. In adults, MSCs are also called resident stem cells and constitute a reserve to replace damaged cells because they are capable of differentiating into a wide variety of cellular types, which is why they have been advocated in tissue-regenerative therapies [146]. MSCs have also been utilized for limb ischemia in both animals and humans with a low level of benefit so far [147,148,149,150]. To improve their potential, some researchers have defended their combination with GH. First, as described, GHR have been found in these progenitor cells [151], facilitating differentiation and growth factor secretion from MSCs by GH. For example, it has been shown that MSCs that overexpress Akt improve their functions, increasing the power to repair damaged myocardium despite infrequent cellular fusion or differentiation [152]. However, MSCs, rather than stimulate growth factor secretion, facilitate the release of growth factors from surrounding tissues [149]. This insight confirms that paracrine mechanisms mediated by MSCs are authentic players in enhancing the survival of existing myocytes and that we could act on cytokines and growth factor secretion by stimulating the Akt pathway. GH might favor the stimulus of this signaling pathway in MSCs. In fact, knockout mice with a GH receptor (GHRKO) have shown how MSCs tend to differentiate into adipocytes, partially losing their potential, both in bone marrow and peripheral tissues. The Wnt/β-catenin signaling pathway seems to increase when GH is present, suggesting that it has a role in the modulation of MSCs’ fate by GH [153]. This fact has also been confirmed in MSCs of human trabecular bones. Human bone marrow MSCs express GHR and respond to GH via JAK2/STAT5 intracellular signaling [154]. These findings support the idea that MSC activity might be modulated with independence of the origin of these cells and that GH can be a true stimulator for them. Thus, since MSCs are essential for vasculogenesis, they could also be crucial during adult arteriogenesis due to the similarity between both processes, and GH could be of help to increase the potential and modulate the action of MSCs or even to facilitate their differentiation in SMCs that favor collateral growth.

### 3.7. GH and Extracellular Matrix (ECM)

Collateral remodeling is the final step for arteriogenesis, and it occurs mainly in the media and adventitia layers. SMCs and fibroblasts play the main role in this phase, in which the external elastic lamina and the adventitia elastin are degraded by proteolytic enzymes such as metalloproteinases (MMP) and plasmin to make room for the growing vessel. FGF-2 or IGF-I, among others, trigger the maturation and proliferation of fibroblasts and SMCs [155]. Cell migration is key for both angiogenesis and arteriogenesis. However, migrating cells need of a scaffold to do so. For this, the extracellular matrix (ECM) plays a pivotal role, and GH can regulate it [156,157]. For instance, when the hormone is administered to cultured human SMCs, it produces a direct and dose-dependent increase of hyaluronic acid and chondroitin [157]. That is, the remodeling process takes place by the dynamic restructuring of the ECM with degradation and synthesis. GH, on the other hand, can act as a mitogenic factor, also favoring the release of growth factors and the migration of a major players in this process such as SMCs, fibroblasts, ECs, and the same macrophages in the vascular wall. In a study with cultured murine thymic endothelioma cells (tEnd.1), treatment with GH for 24 h induced an increase in the production of fibronectin and laminin from these ECs as compared to the control, and it also increased the expression of ECM receptors for fibronectin and laminin, as well as the migratory activity of the aforementioned cell line [156]. On the other hand, the activation of macrophages in the vascular wall facilitate MMP production, influencing ECM degradation. All these data confirm the ability of GH to influence ECM, playing an important role for vessel remodeling.

### 3.8. GH and NO-Independent Vascular Tone: the Role of Sympathetic System

Sympathetic innervation seems to be necessary for stabilizing vascular wall tone and cell phenotype, since in sympathectomized vessels, both SMCs and fibroblasts increase in numbers with collagen alterations (collagen III upregulation and collagen IV downregulation) [158]. This supports the hypothesis that the autonomic system participates in vascular homeostasis. In GHD patients, a marked increase in sympathetic activity has been found [159], but it tends to be reversed after GH replacement therapy [160], suggesting that the hormone may regulate central sympathetic activity, thus affecting vascular peripheral resistance. When sympathetic activity is increased, collaterals suffer an intimal thickening that diminishes SSF and collateral enlargement after ischemia [158]. In skeletal muscle arterioles from diabetic patients with neuropathy, an increased vasomotor tone has been described, while a higher α-adrenergic tone has been found in the iliac artery of diabetic animals [161,162]. Thereupon, proper innervation plays an important role in the development and remodeling of blood vessels, although the exact mechanism of impaired arteriogenesis, when altered, is still poorly understood. 

That GH is related to the autonomous system has not only been described in GHD patients. In healthy humans, sympathetic activation and baroreflex resetting were found after GHRH administration in a microneurographic study of muscle sympathetic nerve activity as compared to a placebo group at rest, whereas blood pressure and heart rate were not altered [163]. Therefore, GH is related to sympathetic system during both the physiological and pathological situations of GH secretion. These findings could help to understand the connections of GH with the vascular system and the role of the hormone in arteriogenesis.

### 3.9. GH and Midkine (MDK) Relation should Be Investigated

Midkine (MK), a heparin-binding growth factor that seems to play an important role in arteriogenesis by mediating eNOS activity and increasing the bioavailability of VEGFA [164], has also been related with GH, as the expression of MK and its receptors have been detected in somatotrophs of both embryonic and adult rats, favoring the development of the pituitary gland and acting as a regulator of its function in adults [165]. MK seems to be secreted by follicle-stimulating cells in the pituitary gland, controlling GH production from GH cells in a paracrine way via protein tyrosine phosphatase receptor-type Z (Ptprz1) [166]. 

In MK-deficient mice, altered vascular remodeling has been seen due to the reduction in ECs proliferation [167]. This means that ECs proliferation plays a pivotal role in correcting vascular growth, and, as described before, GH is mainly involved in EC proliferation, participating, from this point of view, in this mitogenic and proliferative process. 

Given that GH is a major regulator of eNOS expression and NO production in ECs for vasodilation that typically increases VEGF and KDR, and as a consequence of its participation in vascular homeostasis, it is tempting to speculate that both MK and GH might work together to maintain vascular homeostasis, as well as to activate vascular enlargement when an ischemic condition is present. MK could indirectly promote vasodilation by increasing the bioavailability of GH in ECs [168]. However, this hypothesis is not yet proved and needs confirmation.

### 3.10. GH and Klotho: the Perfect Combination?

Of interest is the special relationship between Klotho and GH. A comprehensive review of the relation between GH and Klotho has been previously described [16,169]. Here, the intention is just to highlight the importance of this relationship for arteriogenesis. As is known, Klotho is a type of transmembrane full-length protein mainly found in the kidneys which can act remotely by generating a circulating form (sKlotho) [170]. This form can work as a hormone that regulates the functions of cells that lack this protein, such as ECs and arterial SMCs [171]. It is known that sKlotho increases both GH production by the anterior pituitary [172] and the local production of the hormone by the endothelium. Additionally, sKlotho inhibits the negative feedback of IGF-I on GH secretion [173]. Both molecules are essential for vascular homeostasis, as well as when this process is altered. It is thought that the lack of Klotho production in chronic renal insufficiency is the reason for the aging and calcification of the vessel wall, even in collaterals, and, in consequence, it impairs atherosclerosis and arteriogenesis. In fact, mice overexpressing Klotho have a longer life expectancy due to the mentioned effect of Klotho and its anti-IGF-I action [174]. Thus, Klotho stimulates GH while blocking IGF-I, eliminating the deleterious effects of the latter. However, GH also exerts influences on Klotho levels, as patients with acromegaly show increased levels of Klotho, and they show a downregulation when the GH-produced pituitary adenoma is removed [169].

Klotho seems to be important in physiological states but also in pathological ones, since mice with Klotho deficiency show vascular hyperpermeability. This molecule seems to be necessary for the control of the action of VEGF on the vascular permeability. At the level of the endothelium, Klotho facilitates the association of both VEGFR2 and transient receptor potential canonical calcium channel 1 (TRPC-1) by promoting their cointernalization and regulating calcium entry to maintain homeostasis. Thus, KDR is needed by Klotho to accomplish its actions [175]. GH, as demonstrated before, increases KDR in the ischemic muscle of the leg and could collaborate with Klotho this way by facilitating this action. 

Other important action of Klotho for arteriogenesis is that this factor also participates in the regulation of vascular tone, as it compensates for the vasoconstrictor action of some factors such as phosphates or FGF23 by increasing the production of NO [176], an action probably mediated by local GH. Interestingly, while Klotho favors vascular contraction in aortic ring samples from mice in *in vitro* studies by increasing ROS levels both in SMCs and ECs when is administered alone, the relaxation phenomenon predominates if SMCs are pretreated with FGF23 or phosphates, while ROS levels remain elevated. This is an effect that is mediated by an indirect NO production by Klotho from ECs stimulating both eNOS and iNOS [176]. In fact, when the endothelium is removed, this effect disappears. 

The above data supports again that redox stress contributes to the regulation of vascular homeostasis, since eNOS is sensitive to ROS. It also supports the complex action that Klotho has depending of its environment, as happens with GH. The other important message is the main role of the ECs in the control of vascular tone and SMCs actions, and that Klotho and GH seem to collaborate in physiological and pathological conditions such as ischemia, although this aspect has to be confirmed.

## 4. Conclusions

Vascular homeostasis critically depends on the physiological response of endothelial cells to blood supply and the appropriate redox balance. The endothelium releases many factors to control vascular tone, the adhesion of circulating blood cells, the proliferation of smooth muscle cells, and inflammation. 

Why should GH be considered a promising therapeutic agent for neovascularization? The GH/PRL/PL family regulates the physiological growth and regression of blood vessels in female reproductive organs, and this fact strongly support its vascular role in neovascularization. There is no doubt about the fact that the GH/IGF-I axis has to play an important role in neovascularization, both in physiological and pathological states, as evidence here presented has underlined. This axis suffers an important decline with aging, mainly affecting GH secretion. Considering that most patients with ischemic injuries are elderly, GH therapy could be considered of help in improving vascularization and mitigating symptoms. 

However, information concerning the regulation of neovascularization by proangiogenic hormones such as GH is insufficient, since few physiological or pathological conditions have been deeply studied, with some exceptions. This fact could be explained by the use of different animal models of ischemia, types of tissue analyzed, disease status, hormone doses, or follow-up times. These effects also depend on the relative contribution of the local production of hormones or on the hormonal cleavage by proteases in cells or the clearance of these hormones by kidneys when they are exogenously administered. Surprisingly, data are also limited about endogenously produced antiangiogenic substances that might be overexpressed in chronic states such as ischemia and that could act with a harmful effect on GH actions.

The role of redox balance in arteriogenesis and how GH could aid in the mitigation of it were analyzed. We also proposed the possibility that GH and IGF-I could be parts of those mitogenic factors secreted by endothelial cells in response to shear stress forces. The large number of connections that both molecules have with cytokines, hormones, and cells involved in neovascularization reinforce their role in this process. Finally, in this review, it has been presented some molecular insights from the GHAS trial in patients with critical limb ischemia that correlate perfectly with recent publications on arteriogenesis and that can help to understand the action of GH dealing with ischemia. Nevertheless, the molecular results of this initial clinical study still need to be confirmed in larger studies.

## Figures and Tables

**Figure 1 cells-09-00807-f001:**
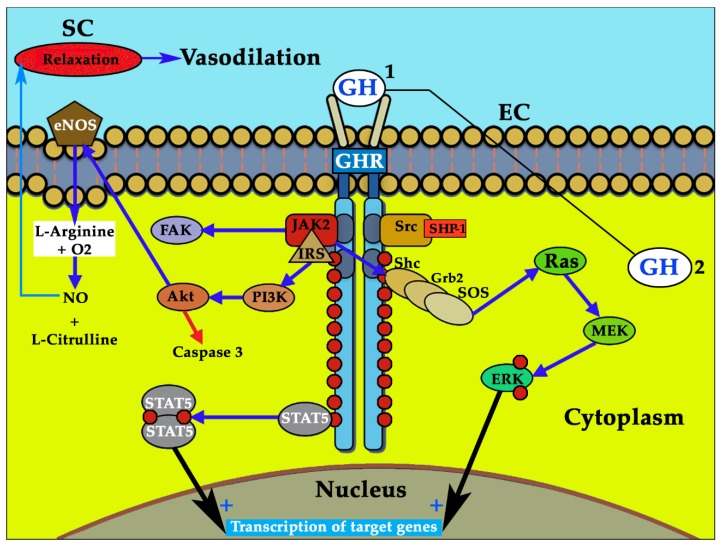
Effects of growth hormone (GH) on the vascular endothelium. The interaction GH–GHR (growth hormone receptor) produces the activation of the associated Janus kinase 2 (JAK2), which induces the phosphorylation (red circles) of tyrosine located in the cytoplasmic receptor domain, leading to the phosphorylation of GH-signaling pathways, such as signal transducers and activators of transcription (STATS). Among STATS (STATS 1 and 3 not shown in the Figure), STAT5 homodimerizes and is translocated to the nucleus, where it induces the transcription of a series of genes. Activated JAK2, acting on the insulin receptor substrate (IRS) induces the phosphorylation of phosphoinositide 3-kinase (PI3K) which, in turn, activates the cell survival factor serine-threonine kinase (Akt). This inhibits the proapoptotic enzyme Caspase 3 (red arrow), but it also activates endothelial nitric oxide synthase (eNOS) (blue arrow). Activated eNOS promotes the synthesis of nitric oxide (NO) (from L-Arginine and O2) and the formation of L-Citrulline. The formed NO flows from the cytoplasm to the muscle cell layer of the blood vessels, producing its relaxation and consequent vasodilation. The interaction GH–GHR also induces the activation of the Shc adapter proteins, which leads to the activation of the Grb2–SOS–Ras–Raf–MEK–ERK (extracellular signal-regulated kinase) pathway (Raf is not shown in the figure). Activated ERK translocates into the nucleus of the endothelial cells (ECs) and regulates the expression of genes involved in cell proliferation, differentiation, and survival, but it also regulates cell motility and migration (key for the formation of new vessels). The GH–GHR interaction also activates focal adhesion kinase (FAK). SHP: protein tyrosine phosphatase. Blue arrows: stimulation. Red arrow: inhibition. Black arrows: Translocation to the nucleus. 1: endocrine GH. 2: endothelial GH: plays and auto/paracrine role and in situations of the absence of endocrine GH perhaps plays the role of the former (black line).

**Figure 2 cells-09-00807-f002:**
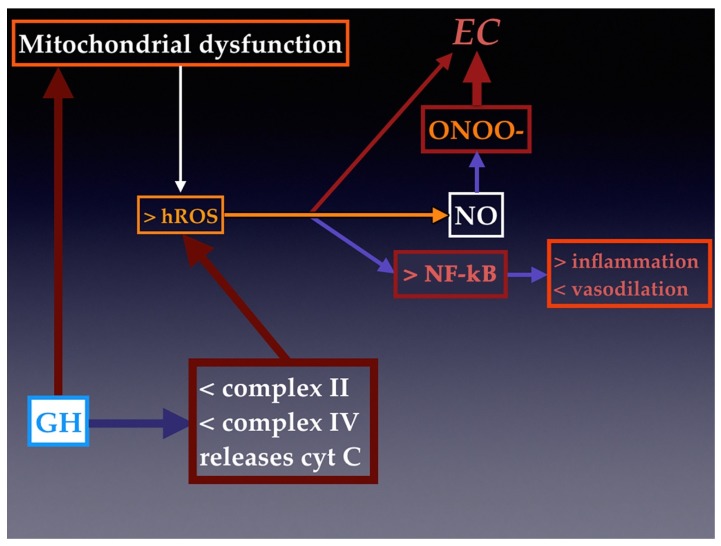
Mitochondrial dysfunction leads to vascular endothelium senescence. An excessive production of highly reactive oxygen species (hROS) by mitochondria induces the elimination of NO, because it is transformed into peroxynitrite (ONOO^–^), which is toxic for endothelial cells. Moreover, the excessive oxidative stress induces the activation of nuclear factor-kappa B (NF-kB), which increases the expression of inflammatory genes in the blood vessels and decreases vasodilation. These lead to the presentation of a senescent phenotype in the endothelial cells. GH administration corrects mitochondrial dysfunction, because GH is able to enter to mitochondria and decrease the activity of complexes II and IV of the mitochondrial respiratory chain. In addition, GH produces a decrease in the mitochondrial membrane potential which translates into the release of cytochrome C (cyt C) to the cytosol. Blue arrows: stimulation. Red arrows: inhibition or damage (in endothelial cells). Orange arrow: indicates that an increase in hROS induces the transformation of NO into ONOO^-^.

**Figure 3 cells-09-00807-f003:**
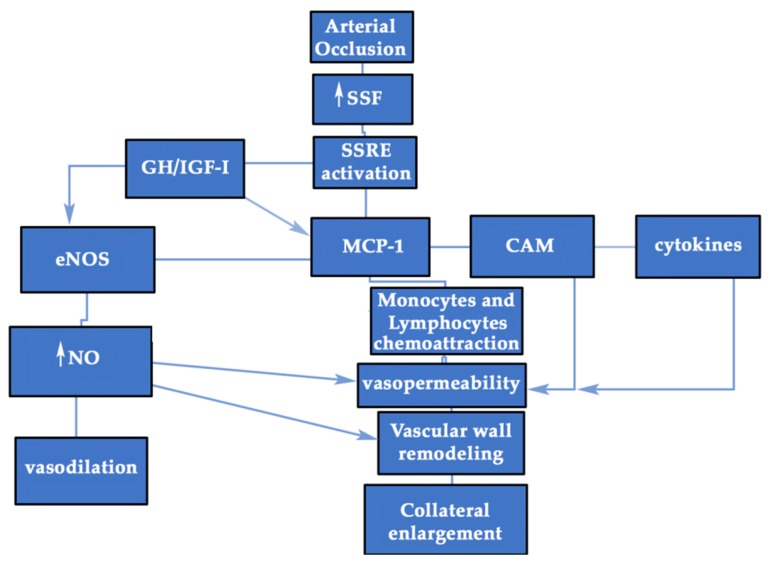
GH/IGF-I are also produced in response to shear stress forces (SSF) for collateral enlargement. Schematic representation of the activation of several molecules after the increase in SSF in which GH and IGF-1 have been added as new elements that enhance these mechanisms. Since GH and IGF-1 are potent mitogenic agents, they have to play a role in the translation of mechanosensing signals. The NO pathway not only is important for vasodilation, as it also has many actions in chemoattraction of inflammatory cells and vasopermeability. The local production of cytokines and hormones seems to be essential for collateral enlargement. SSF: shear stress forces; SSRE: shear stress response elements; eNOS: endothelial nitric oxide synthase; MCP-1: monocyte chemoattractant protein-1; CAM: cellular adhesion molecules; NO: nitric oxide.

**Figure 4 cells-09-00807-f004:**
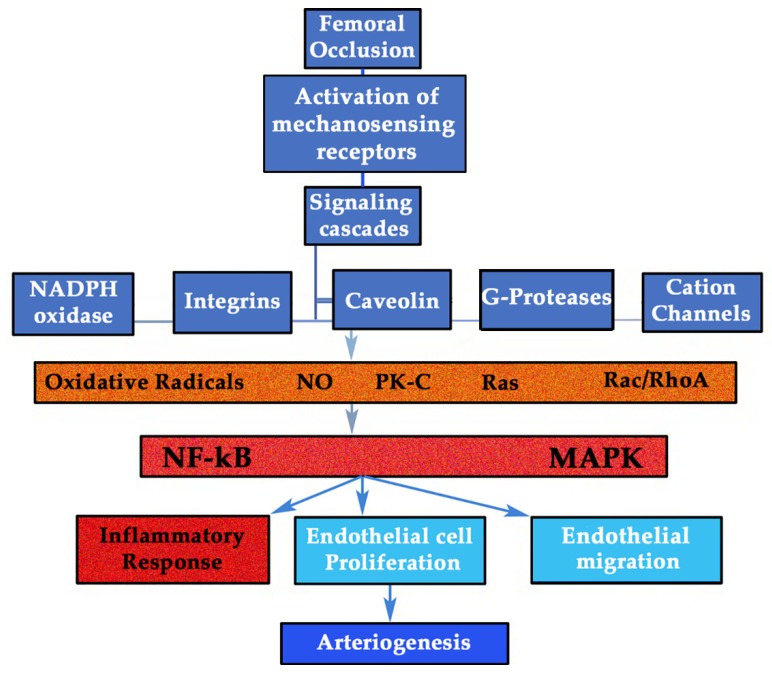
Signaling cascades for collateral growth: The redox system can play a very important role. Schematic representation of signaling cascades after an arterial occlusion, triggering both the inflammatory response and the phenomena of the proliferation and migration of endothelial cells into the artery media layer. Several pathways have been highlighted: Rho, involved in endothelial cells proliferation; Ras, for endothelial cells migration; and NO, for endothelial function and monocytes adhesion. The oxidative radicals produced in cellular metabolism are currently considered very important in the stimulation of the NO pathway for vascular homeostasis. They control NO bioavailability. The Rho pathway has been advocated as crucial for sensing of SSF. Caveolins: family of integral membrane proteins that play a role in the signaling of integrins and in the migration of endothelial cells. Cation channels, mainly the Ca^2+^ ion, are also related to the activation of Protein kinase C (PKC) and the RAS/RAC (rats’ sarcoma-extracellular signal-regulated kinases/Ras-related C3 botulinum toxin substrate). All signaling cascades are activated by SSF. For more details, see reference [106,107]. NADPH oxidase: nicotinamide adenine dinucleotide phosphate oxidase; NO: nitric oxide; PK-C: protein kinase C, Ras: rats’ sarcoma-extracellular signal-regulated kinases (currently known as GTPase Ras); Rho, hexameric protein found in prokaryotes, necessary for the process of terminating the transcription of some genes, Rac: Ras-related C3 botulinum toxin substrate (subfamily of the Rho family); NF-kB: nuclear factor-kappa B. MAPK: mitogen-activated protein kinase.

**Figure 5 cells-09-00807-f005:**
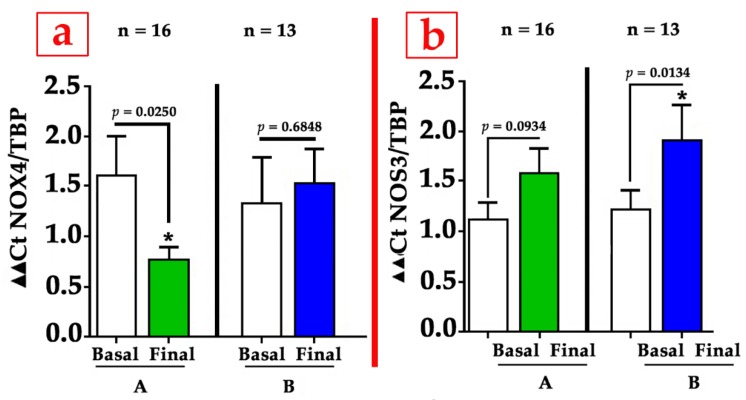
GH decreases the redox stress in ischemic muscles from the calf of patients with critical limb ischemia. Insights from the GHAS trial. (**a**)**:** shows the significant decrease in NOX4 mRNA levels only seen in the GH-treated group (**A,** green bars) and not in Placebo-treated group (**B,** blue bars). (**b**): depicts the significant increase in the levels of NOS3 (eNOS) mRNA during the period of treatment in patients with placebo related to its baseline levels, while in the GH group (**A**), although there was also an increase in NOS3 mRNA levels after the treatment, this increase was not significant. Samples from soleus muscle. Group A: GH; Group B: placebo. Basal: baseline mRNA levels before treatment; final: final mRNA levels after 8 weeks of treatment. Statistics: non-parametric test. Note that of the 36 patients initially recruited for the study, complete muscle samples (basal and final) were only obtained from 28 of them as a result of deaths, limb amputations, or the patient´s refusal to allow for the second biopsy. * indicates statistically significant.

**Figure 6 cells-09-00807-f006:**
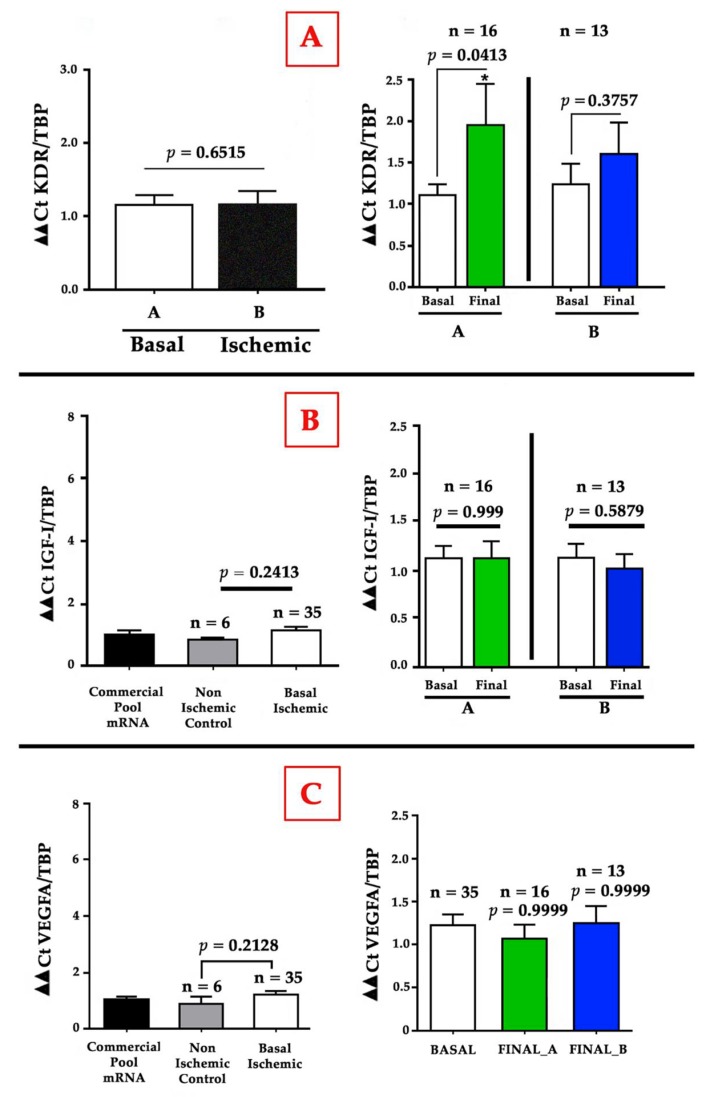
GH increases VEFGA-R2 or KDR/Flk-1 mRNA levels in ischemic muscles of patients with critical limb ischemia without apparent changes in IGF-I and vascular endothelial growth factor A (VEGFA) mRNA levels. Insights from the GHAS trial. **(A).** VEFG-R2 or KDR/Flk-1 mRNA levels in the GHAS trial. The left graph shows the baseline levels between both groups of treatment with no significant differences. In the right graph, a significant increase in KDR mRNA levels in the GH group compared to the placebo group were found during the period of treatment (8 weeks). **(B).** IGF-I mRNA levels in the GHAS trial. The graph on the left shows levels of IGF-I at baseline in ischemic vs non-ischemic muscle samples with no differences. The graph on the right shows the lack of changes in any group of treatment during the period of treatment. **(C).** VEGFA mRNA levels in the GHAS trial. The left graph depicts the comparation between VEGFA levels in ischemic vs non-ischemic muscle samples in the calf at baseline. The right graph depicts the lack of significant changes after 8 weeks of treatment. Group A, green bars: GH; Group B, blue bars: placebo. Non-ischemic muscle samples: sample of reference obtained from amputations in patients without limb ischemia. Commercial pool mRNA: a commercial mRNA from skeletal muscle used as a technical control for normalization of the different assays. A non-parametric test was used for the statistics. Note that although at the beginning of the study there were data from 35 muscle samples, at the end of this study, there were only complete data from 28 patients (basal and final samples). * indicates statistically significant.

**Table 1 cells-09-00807-t001:** Main characteristics of the patients recruited in the GHAS trial. Enrolled patients: patients with informed consent; age: years old ± standard deviation; ABI: ankle–brachial index; CRF: chronic renal failure; DM: diabetes mellitus; and rhGH: recombinant human growth hormone.

Enrolled Patients	Sex	Age (Mean ± SD)	Comorbidity	Baseline ABI (Mean)	Rutherford class 5–6
36	Male: 28	71 ± 12.7	Heart disease:47%	0.19	70.6%
	Female:8		DM:59%		
			Neuropathy:57.6%		
			CRF:26.5%		
**Therapy**	**Dose**	**Duration of treatment**	**Follow-up**	**Follow-up Periods**	
rhGH vs. Placebo	0.4 mg/day	8 weeks	12 months	0–2; 3–6; 7–12 months	

**Table 2 cells-09-00807-t002:** Specific characteristics of the patients distributed by treatment groups studied in the GHAS trial. A homogeneous distribution can be seen, except for tobacco consumption. HT: arterial hypertension; DM: diabetes mellitus; CRF: chronic renal failure; HD: heart disease; *n* = sample size; y = year. Note that in this table, we show data of 34 of 36 patients enrolled, because two patients died after the signing of the informed consent.

		Total		GH Group		Placebo Group		*p* Value
		n	%	n	%	n	%	*p*
Age	< 65 65–80 > 80	14 8 12	41.18 23.53 35.29	6 3 9	33.33 16.67 50	8 5 3	50 31.25 18.75	0.1211
Gender	Male Female	27 7	79.41 20.59	13 5	72.22 27.78	14 2	87.5 12.5	0.2715
Etiology	Atherosclerosis Buerger Scleroderma Mix	24 3 1 6	70.59 8.82 2.94 17.65	11 1 1 5	61.11 5.56 5.56 27.78	13 2 - 1	81.25 12.5 - 6.25	
HT	No Yes	12 22	35.29 64.71	4 14	22.22 77.78	8 8	50 50	0.0907
DM	No Yes	14 20	41.18 58.82	6 12	33.33 66.67	8 8	50 50	0.3243
CRF	No Yes	25 9	73.53 26.47	13 5	72.22 27.78	12 4	75 25	0.8546
HD	No Yes	18 16	52.94 47.06	8 10	44.44 55.56	10 6	62.5 37.5	0.2924
Dialysis	No Yes	33 1	97.06 2.94	17 1	94.44 5.56	16 -	100 -	0.3386
Tobacco	No Ex-smoker <1y Smoker	24 2 8	70.59 5.88 23.53	16 1 1	88.89 5.56 5.56	8 1 7	50 6.25 43.75	0.0107
Rutherford	3 4 5 6	5 5 15 9	14.71 14.71 44.12 26.47	3 - 9 6	16.67 - 50 33.33	2 5 6 3	12.5 31.25 37.5 18.75	0.188
Rest Pain	No Yes	8 26	23.53 76.47	5 13	27.78 72.22	3 13	18.75 81.25	0.5356
Trophic Lesion	No Yes	11 23	32.35 67.65	3 15	16.67 83.33	8 3	50 50	0.0381
Neuropathy	No Yes	14 19	42.42 57.58	7 10	41.18 58.82	7 9	43.75 56.25	0.8812

**Table 3 cells-09-00807-t003:** GH decreases excess inflammation by lowering the circulating levels of TNF-α in patients with critical limb ischemia. Insights from the GHAS trial. Evolution of plasmatic levels of TNF-α and C reactive protein (CRP) in the GHAS trial after 8 weeks of treatment. TNF-α was significantly elevated in patients treated with GH compared to the placebo at baseline, which means that severe ischemia is accompanied by an increase in inflammation. In addition, inflammation was more severe in the group treated with GH, which was seen by the fact that levels of both TNF-α and CRP were significantly elevated at baseline in the GH group as compared to the placebo group. However, after 8 weeks of treatment, only patients receiving GH showed a significant decrease of circulating levels of TNF-α (TNF-α final), while CRP levels did not significantly change compared to baseline levels. TNF-α baseline: TNF-α levels at baseline (time before administering the treatment); TNF-α final: TNF-α levels after 8 weeks of treatment. Determination of TNF-α plasmatic levels: ELISA test (Quantikine, R&D Systems). Normal reference value for TNF-α levels: <8.1 pg/mL. Obs.: number of patients analyzed. SD: standard deviation. CRP: C-reactive protein (mg/L), normal values: < 1 mg/L. Note that only 32 of 36 patients in the GHAS trial had available baseline data of these markers. At the end of the treatment, there were some missed samples as a consequence of deaths or patient´s refusal to obtain the blood sample.

	GH-Treated			Placebo			
	Obs.	Mean	SD	Obs.	Mean	SD	*p* Value
TNF -α Baseline	16	12.35	5.2	16	8.78	3.94	0.018
TNF -α Final	15	10.92	5.12	14	8.04	3.6	0.046
CRP Baseline	16	2.07	2.86	16	0.78	0.69	0.045
CRP Final	15	1.1	1.34	14	3.42	7.51	0.218

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
