# Peer review of "Why Should Growth Hormone (GH) Be Considered a Promising Therapeutic Agent for Arteriogenesis? Insights from the GHAS Trial"

_cells, 2020, doi:10.3390/cells9040807_

Round 1
Reviewer 1 Report
It is interesting and intriguing review article. After reading this article submitted to me for review, however, it occurred to me observations and comments. The described comments and suggested changes in the text lead to a better understanding of the theme and will increase readers' interest in this topic. Here they are.
Unfortunately, the authors a bit spontaneously, in my opinion too briefly described the physiological role of GH and omitted the topic of IGF-1 and its role in the organism. The authors wrote: „The persistent GH secretion out of the growth period is a clear proof of the importance of the actions of this hormone at multiple levels as the cardiovascular, hematopoietic and immune systems, among others [1].” The fact that this article primarily focuses on the answer to the question why should growth hormone be considered a promising therapeutic agent for arteriogenesis, it can not be the basis for not raising this topic.
Authors wrote: “A recent study in subjects without GH-deficiency (GHD) or any cardiovascular disease (CVD), but with one or more CV risk factors (age, smoking, obesity, hypertension, dyslipidemia, insulin resistance), demonstrated that GH and its mediator IGF-I play a protective role in arterial wall changes associated with vascular aging [6].” This also requires development and expansion. The hGH gene family is composed by two growth hormone (GH) genes (GH-N and GH-V), and three placental genes that are located in the chromosome 17 [PMID: 3030680]. In the case of GH-N, it is already well known that in addition to its pituitary expression, which is responsible for the actions of the hormone at the endocrine level, the hormone is also expressed in numerous cells and tissues, where it acts in an auto/paracrine manner (PMID: 27773998). Mechanistically, GH responsiveness and the GH–IGF-1-axis are subject to regulation, both in terms of negative feedback on GH and IGF-1 secretion, but also in terms of GH receptor (GHR) and IGF-1 receptor (IGF1R) signalling (PMID: 28946616).
The regulation of GH pituitary expression is very complex, since in the last few years the classical knowledge of a positive regulation by GHRH and negative by somatostatin, has been changed after the knowledge of a series of factors that are decisively involved in that regulation (PMID: 29346331). This is the case, for instance, of the orexigenic Ghrelin, released by the empty stomach, or the postulated anti-senescence factor Klotho, mainly expressed in the kidney, but also in the brain and in the own somatotroph cells where it would act in an auto/paracrine manner for directly regulating GH secretion (PMID:271257397). The many factors, inhibits GH-induced hepatic expression of Insulin-like Growth Factor I (IGF-I), therefore inhibiting the IGF-I effect on hypothalamic somatostatin release and the direct negative effect of IGF-I on pituitary somatotrophs, thus acting as a coordinator between body growth or other IGF-I dependent GH effects on the human body (PMID: 28572090, PMID: 29346331).
In the text of manuscript, the authors should write necessarily some words about IGF and related factors on organogenesis and tissue regeneration, especially in the context of arteriogenesis. As the authors wrote, therefore better understanding of the role GH as therapeutic agent for arteriogenesis may certainly contribute to the development of new more effective and safer therapies. For example, during postnatal period of life, endogenous and exogenous IGF-1 was shown to promote the regeneration of different tissues, including the bone (PMID: 1570765), muscle (PMD: 1442129), nerve (PMID: 11396586) and pancreas (PMID: 14726612). These pro-regenerative effects of IGF-1 are additionally associated with reduction in the release of pro-inflammatory cytokines and stimulation of the release of anti-inflammatory cytokines (PMID: 14726612). Also ghrelin, which activates the hormonal axis: growth hormone-IGF-1 and increases the release of endogenous IGF-1 (PMID: 25716961; PMID: 17033095), exhibits protective and therapeutic effects in different tissues, for example: heart (PMID: 14556085), spinal cord (PMID: 22949835), kidney (PMID: 16306169), pancreas (PMID: 25594510; PMID: 20814069; PMID: 21081804), colon (PMID: 27598133, PMID: 28538694) and oral mucosa (PMID: 29151078). This information is extremely important and requires adding, especially in the context of reports that were presented by the authors.
This topic requires adding and briefly discussing the problem. Such a short extension of this topic will undoubtedly raise the quality of this manuscript.
Author Response
Comments and Suggestions for Authors
It is interesting and intriguing review article. After reading this article submitted to me for review, however, it occurred to me observations and comments. The described comments and suggested changes in the text lead to a better understanding of the theme and will increase readers' interest in this topic. Here they are.
Unfortunately, the authors a bit spontaneously, in my opinion too briefly described the physiological role of GH and omitted the topic of IGF-1 and its role in the organism. The authors wrote: „The persistent GH secretion out of the growth period is a clear proof of the importance of the actions of this hormone at multiple levels as the cardiovascular, hematopoietic and immune systems, among others [1].” The fact that this article primarily focuses on the answer to the question why should growth hormone be considered a promising therapeutic agent for arteriogenesis, it cannot be the basis for not raising this topic.
The aforementioned paragraph has been modified according to the reviewer suggestion. However, the aim of this manuscript is to review the possible role of GH in arteriogenesis and to offer same molecular data about whether GH administration may be useful in a real scenario for patients with critical limb ischemia with no other alternative than the amputation of the limb. In this sense, we obtained very good results in several patients so treated, although this has not been published yet. Thus, these molecular data represent a novelty for this special issue of Cells. Therefore, it seems to us that introduce more aspects about the multiple effects of GH in the human body would only repeat our previous publication (reference 1). Regarding IGF-I, its role on vascular homeostasis has also been widely analyzed through the manuscript.
Authors wrote: “A recent study in subjects without GH-deficiency (GHD) or any cardiovascular disease (CVD), but with one or more CV risk factors (age, smoking, obesity, hypertension, dyslipidemia, insulin resistance), demonstrated that GH and its mediator IGF-I play a protective role in arterial wall changes associated with vascular aging [6].” This also requires development and expansion.
This expression is only an anticipation of the extensive analysis we carry out throughout the manuscript. We do not believe that it is necessary to explain this concept in more detail at this point, as it is discussed in detail later in the manuscript.
The hGH gene family is composed by two growth hormone (GH) genes (GH-N and GH-V), and three placental genes that are located in the chromosome 17 [PMID: 3030680]. In the case of GH-N, it is already well known that in addition to its pituitary expression, which is responsible for the actions of the hormone at the endocrine level, the hormone is also expressed in numerous cells and tissues, where it acts in an auto/paracrine manner (PMID: 27773998). Mechanistically, GH responsiveness and the GH–IGF-1-axis are subject to regulation, both in terms of negative feedback on GH and IGF-1 secretion, but also in terms of GH receptor (GHR) and IGF-1 receptor (IGF1R) signalling (PMID: 28946616).
You are right in your description, but once again we do not think it is important in this review to add any description on how GH is regulated and how GH family genes are expressed in the pituitary and the placenta, since we are describing how GH / IGF-I axis acts in vessels, so that GH may be administered as therapy to patients with critical limb ischemia, elderly patients who do not have pituitary secretion of the hormone, since it progressively decreases from 20 years until being practically absent from 50 years. On the other side, the description of the expression of GH in practically all organs and tissues and its auto/paracrine effects have been described in the manuscript.
The regulation of GH pituitary expression is very complex, since in the last few years the classical knowledge of a positive regulation by GHRH and negative by somatostatin, has been changed after the knowledge of a series of factors that are decisively involved in that regulation (PMID: 29346331). This is the case, for instance, of the orexigenic Ghrelin, released by the empty stomach, or the postulated anti-senescence factor Klotho, mainly expressed in the kidney, but also in the brain and in the own somatotroph cells where it would act in an auto/paracrine manner for directly regulating GH secretion (PMID:271257397). The many factors, inhibits GH-induced hepatic expression of Insulin-like Growth Factor I (IGF-I), therefore inhibiting the IGF-I effect on hypothalamic somatostatin release and the direct negative effect of IGF-I on pituitary somatotrophs, thus acting as a coordinator between body growth or other IGF-I dependent GH effects on the human body (PMID: 28572090, PMID: 29346331).
Yes, we agree with you, but we only try to describe the effects of exogenously administered GH as a therapy for critical limb ischemia in elder patients.
In the text of manuscript, the authors should write necessarily some words about IGF and related factors on organogenesis and tissue regeneration, especially in the context of arteriogenesis. As the authors wrote, therefore better understanding of the role GH as therapeutic agent for arteriogenesis may certainly contribute to the development of new more effective and safer therapies.
We already wrote about IGF-I effects on arteriogenesis. The effects of IGF-I on other tissues and organs are not the object of this manuscript.
For example, during postnatal period of life, endogenous and exogenous IGF-1 was shown to promote the regeneration of different tissues, including the bone (PMID: 1570765), muscle (PMD: 1442129), nerve (PMID: 11396586) and pancreas (PMID: 14726612).
We know it, but again these effects of IGF-I are not the object of this manuscript.
These pro-regenerative effects of IGF-1 are additionally associated with reduction in the release of pro-inflammatory cytokines and stimulation of the release of anti-inflammatory cytokines (PMID: 14726612). Also ghrelin, which activates the hormonal axis: growth hormone-IGF-1 and increases the release of endogenous IGF-1 (PMID: 25716961; PMID: 17033095), exhibits protective and therapeutic effects in different tissues, for example: heart (PMID: 14556085), spinal cord (PMID: 22949835), kidney (PMID: 16306169), pancreas (PMID: 25594510; PMID: 20814069; PMID: 21081804), colon (PMID: 27598133, PMID: 28538694) and oral mucosa (PMID: 29151078). This information is extremely important and requires adding, especially in the context of reports that were presented by the authors.
We added the description of the IGF-I effects on the reduction in the release of pro-inflammatory cytokines and the stimulation of the release of anti-inflammatory cytokines. The effects of IGF-I on kidneys, spinal cord, colon and oral mucosa, are not the object of this manuscript. The effect of ghrelin on eNOS expression had been described in the text.
This topic requires adding and briefly discussing the problem. Such a short extension of this topic will undoubtedly raise the quality of this manuscript.
Reviewer 2 Report
In this review, the authors argue for using GH for the treatment of limb ischemia. The manuscript appears to mix elements of a review and original research. Indeed, the title does not seem to cover the topic. There are a number of concerns. A few examples:
The review part of the manuscript lacks novelty for the mechanisms of arteriogenesis have received considerable attention and were subject to numerous reviews. Many of the authors’ interpretations can be argued. This part of the manuscript can easily be shortened by 60% without affecting the message.
The original research, the GHAS trial lacks proper description of patient characteristics (age, sex, co-morbidities and medications just to mention a few).
Figure 2. If red arrows represent inhibition, how would hROS reduced NO production would lead to enhanced ONOO- formation?
NF-kB has been shown to regulate transcription of genes (e.g. the I-kB gene) that are involved in the termination of inflammation. How this would fit with GH actions? NF-kB is misspelled on the figure.
While GH may bind to mitochondria, it is uncertain whether this could happen in vivo. Is there any evidence that GH decreased mitochondrial ROS production in vivo? What are the mechanisms (receptors)? How would GH affect mitochondria if GHR is expressed on the cell surface?
What is the relevance of discussing GH effects in hematological malignancies to critical limb ischemia?
Figure 3. Do the authors suggest that NO produced by eNOS functions as a chemoattractant for inflammatory cells? How would vasodilatation lead to “collateral enlargement”?
The authors argue that locally or centrally produced GH regulates “gene expression in a different way” without providing any explanation.
How GHR gene expression is regulated. What does “nutritional intake” or medications refer to?
If B cells are major source of GH in blood, what is their relation to collateral enlargement? What evidence indicates that GH could selectively recruit immune cells into the vessel wall, and then locally produced GH would activate these cells to produce cytokines? Figure 4 does not even mention these cell types.
Some description of patient characteristics enrolled in the GHAS trial would be helpful.
What do “monocytes have traditionally been reduced to immune actions” or “insensibility to NO” mean?
Table 1. Group A should be labeled as GH-treated , Group B as placebo. What do the numbers 0 and 2 mean after TNF-alpha?
Figure 6. If non-parametric tests were employed why was the Kolmogorov-Smirnov test performed?
Figure 7. What conclusion can be drawn at n=2 or 3 for a clinical trial?
If the relation between GH and Klotho “has been perfectly described”, what was the rationale for repeating this?
There are several unusual terms, e.g. “vein cava” should read “vena cava”.
Author Response
Comments and Suggestions for Authors
In this review, the authors argue for using GH for the treatment of limb ischemia. The manuscript appears to mix elements of a review and original research. Indeed, the title does not seem to cover the topic.
Although it is true that we mix review and original data research in the text, we believe that the title fits perfectly with the manuscript. The original idea in this special issue on Arteriogenesis and Therapeutic Neovascularization for the journal Cells was just to make a comprehensive review on those aspects that relate GH exclusively to arteriogenesis in order to focus the theme. As known, GH may have many therapeutic possibilities, but, in this field, this hormone has to deal for example, with the arterial growth that takes place, physiologically, every month in females during their reproductive lives. Despite this fact, only a few studies address this issue, even though it is recommendable to mimic the physiologic process of arteriogenesis when an ischemic condition is approached. This was the reason for presenting our first molecular data obtained in a real clinical scenario. We thought that both this journal and review could be the perfect environment to do it. Besides, we wanted to add some insights about the real role of redox balance for improving arteriogenesis.
There are a number of concerns. A few examples:
The review part of the manuscript lacks novelty for the mechanisms of arteriogenesis that have received considerable attention and were subject to numerous reviews. Many of the authors’ interpretations can be argued. This part of the manuscript can easily be shortened by 60% without affecting the message.
Thanks for this commentary. We believe that our revision brings novelties that had been overlooked, as the role of the shear stress in GH/IGF-1 production. These articles, although not new, have never been used to justified the possible role of GH/IGF-1 in arteriogenesis, at least in our knowledge. Given the extremely important role of shear stress for triggering the arteriogenic process, and given that the organism physiologically stimulates this pathway, it is something that encouraged us to perform this review. The extension has been also adjusted for a better understanding and support of what we wanted to say and demonstrate. Thus, in our opinion, this part should not be shortened.
The original research, the GHAS trial lacks a proper description of patient characteristics (age, sex, co-morbidities, and medications just to mention a few).
It was not the objective to present here the GHAS trial description or patients characteristics, that will be presented further. As it was mentioned, this special issue of Cells focuses on molecular mechanisms and not on clinical data. Details of the GHAS trial can be consulted. GHAS trial: Eudract 2012- 002228-34, approved by the Spanish Agency of Drugs and Health Products (AEMPs) and the Autonomic Committee on Research Ethics in Galicia (CAEIG), Spain. However, these characteristics are now shown in Table 2.
Figure 2. If red arrows represent inhibition, how would hROS reduced NO production would lead to enhanced ONOO- formation?
This had been explained in Figure 2. However, we changed now the red arrow from hROS to NO by an orange one. This orange line indicates that the increase in hROS occurring in mitochondrial dysfunction leads to the transformation of NO in ONOO-, therefore decreasing the availability of NO and affecting Endothelial cells.
NF-kB has been shown to regulate transcription of genes (e.g. the I-kB gene) that are involved in the termination of inflammation. How this would fit with GH actions? NF-kB is misspelled on the figure.
NF-kB has been corrected in figure 2 and in the legend of this Figure. It was a mistake. With regard to its actions and activation, it depends on many factors, among them hROS. It is true that I-kBs are inhibitors of kB, allowing NF-kB to enter into the nucleus where it activates the transcription of different genes involved in inflammatory responses or immune responses, and also cell survival or cell proliferation. Therefore these I-kBs proteins terminate the inflammatory state. However, it is also true that NF-kB also activates I-kBs, establishing a kind of automatic feedback cycle between them. But the real thing is that the acute response to hROs is the activation of NF-kB and the subsequent inflammatory response.
While GH may bind to mitochondria, it is uncertain whether this could happen in vivo. Is there any evidence that GH decreased mitochondrial ROS production in vivo? What are the mechanisms (receptors)? How would GH affect mitochondria if GHR is expressed on the cell surface?
In references 48 and 49 of the previous manuscript (49 and 50, respectively, in the revised version), we explain that both GH and its receptor can be internalized to the mitochondria through a new pathway constituted by caveolae. Moreover, in that study (49) the authors found that GH and GHR bind to mitochondrial membranes and GH has been found in mitochondrial fractions early after its injection to rats. These authors also showed that in CHO cells expressing GHR GH exerts an effect on mitochondrial function. They concluded that “GH is specifically imported in mitochondria, where it operates a direct metabolic effect, independently of cell surface receptors and signal transduction” (Reference 49 (old), 50 (new)). The fact that GHR is expressed on the cell membranes does not mean any significant thing, since it is well known that after interacting with the membrane receptor, both GH and its receptor can be internalized and translocated to the nucleus, where GHR acts, and perhaps GH too, as a transcription factor. Moreover, studies in human cortical brain cells pretreated with methadone demonstrated that GH recovers mitochondrial function after methadone-induced toxicity (Nylander E. et al. Int J Mol Sci 2018, 19. pii:E3627. Data not shown). As far as we know there are not data showing the GH effect on hROS production in vivo, but this is a good idea to be studied. We added this to the manuscript.
What is the relevance of discussing GH effects in hematological malignancies to critical limb ischemia?
In this section, we are analyzing the effects of GH on EPCs, particularly CD34+ cells useful for repairing blood vessels and increasing the vascularization of ischemic tissues. Therefore, we described some situations in which GH induces the production and release of these cells. Among these situations, we believe that the effect of GH on this bone marrow production of EPCs (CD34+ in special) in patients who had undergone myeloablative therapy is of particular interest given their situation compared to normal subjects or GHD patients. It is only an example of the effects of the hormone on EPCs production by the bone marrow, without talking about critical limb ischemia.
Figure 3. Do the authors suggest that NO produced by eNOS functions as a chemoattractant for inflammatory cells? How would vasodilatation lead to “collateral enlargement”?
Thanks for this consideration.
About the first question, effectively, endothelium-derived NO is an important vasodilator, an inhibitor of platelet adhesion and aggregation, an inhibitor of monocyte adhesion and inhibitor of vascular smooth muscle cell growth, being considered as an endogenous anti-atherosclerotic molecule. We just wanted to highlight the fact that NO enhances the entry of monocytes into the vascular wall. Now, in Figure 3 this arrow has been erased because it could generate confusion.
About the second, as it is known, the first phase for compensation after an arterial occlusion is the vasodilation of the collateral network. Although essential for compensation, it is not essential for collateral enlargement. Nevertheless, the increase in flow through collaterals leads to an increase in the number of cells and elements necessary for arteriogenesis. Besides, vasodilation by NO traduces an SMC activation, probably contributing to collateral enlargement. According to your indications, we decided to erase in Figure 3 this blue line from vasodilation to collateral enlargement.
The authors argue that locally or centrally produced GH regulates “gene expression in a different way” without providing any explanation.
The explanation for the differential effects of endocrine and autocrine GH had been described many years ago. This was the reason by which we didn’t introduce it in the manuscript. Now two references and explanations have been included.
How GHR gene expression is regulated. What does “nutritional intake” or medications refer to?
GHR gene expression is regulated by many different factors, including GH itself; in the case of ECs it seems that HIF-1alpha, produced in response to hypoxia, is the main factor responsible for the expression of GHR. This has been explained in the text now. Regarding nutritional intake, it has been described as an important factor in inducing GHR expression. This is logical, since GH is a lipolytic hormone, and when fat tissue mass increases, such as in obesity, a reduction in O2 diffusion into cells occurs. This leads to an increase in HIF-1alpha which, in turn, increases the expression of GHR so that GH may act and inducing lipolysis. The opposite situation can be seen in Anorexia Nervosa patients. Contrariwise, TNFalpha treatments reduce GHR expression, at least in cell cultures. Insulin also affects GHR sensitivity.
If B cells are major source of GH in blood, what is their relation to collateral enlargement? What evidence indicates that GH could selectively recruit immune cells into the vessel wall, and then locally produced GH would activate these cells to produce cytokines? Figure 4 does not even mention these cell types.
Currently knowledge about immune cells in arteriogenesis focus in Monocytes, T-lymphocytes and NK cells, with no apparent room for B-lymphocytes. We just wanted to justified that immune cells have receptors for GH and that these cells produce the hormone. This implies that when they are needed in some circumstances, such as arteriogenesis after arterial occlusion, GH could be one of the produced factors participating, and that administering GH we could potentiate this phenomenon. There is no experimental data, as far as we know, supporting this fact, being just a hypothesis. Nevertheless, as it has been demonstrated (see: Hattori, N. Expression, Regulation and Biological Actions of Growth Hormone (GH) and Ghrelin in the Immune System. Growth Horm. IGF Res. 2009, 19 (3), 187–197. https://doi.org/10.1016/j.ghir.2008.12.001), GH enhances monocytes migration in the direction of the produced cytokines, in this case, from collateral arteries. And even more, cytokines and mitogens facilitate GH production from the immune cells. Thus, although the mechanisms is not described, GH is a necessary actor.
To not create confusion, the sentence “B cells are a major source of GH in blood”, has been removed from the text.
Some description of patient characteristics enrolled in the GHAS trial would be helpful.
These are now shown in a new Table (Table 2), although the molecular studies carried out have not finished and they will be published in a near future.
What do “monocytes have traditionally been reduced to immune actions” or “insensibility to NO” mean?
Both expressions have clearly been defined in the text. We wanted to express that monocyte are not cells only involved in defending our organisms conversing into macrophages and phagocyting cells, but they can participate actively in many processes producing important cytokines. The expression insensitivity to NO was coined by Dr. Unthank (see: Unthank, J.; Haas, T. L.; Miller, S. Impact of Shear Level and Cardiovascular Risk Factors on Bioavailable Nitric Oxide and Outward Vascular Remodeling in Mesenteric Arteries. Arter. Mol. Regul. Pathophysiology Ther. I 2011, No. May 2014, 89–119. ) referred to the fact that in ischemic patients rather than a depletion in NO there is a situation in which this NO is rapidly inactivated by ROS, and, so, the artery does not respond to it. To a better understanding, this expression has been changed to “NO-insensitivity”
Table 1. Group A should be labeled as GH-treated, Group B as placebo. What do the numbers 0 and 2 mean after TNF-alpha?
Thanks for these clarifications. They have been added to the text.
0: baseline: the time before administering the treatment; 2: after two months of treatment.
Figure 6. If non-parametric tests were employed why was the Kolmogorov-Smirnov test performed?
The K-S test was employed first to the application of the non-parametric analysis to check the probability to use a parametric test. In statistics, we always check the possibility of using a parametric test seeing the normal distribution of the sample. If this condition is not present, then we use a non-parametric test as U-Mann Whitney. Besides, we have also used a commercial pool of reference, comparing our sample with a reference probability distribution (one-sample K–S test). Unlike it occurs with the U-Mann Whitney test, that compares to population means, K-S is sensitive to any difference between two distribution (symmetry, variability, central tendency, etc), and thus, it could be used as a control of quality or to check the goodness of the fit test.
Figure 7. What conclusion can be drawn at n=2 or 3 for a clinical trial?
No conclusion can be drawn, and it is highlighted in the text. It is simply something that supports the fact that GH could increase these cells in patients with critical limb ischemia, beyond that in experimental studies and healthy volunteers. The final results of the trial will be obtained when the molecular analysis being performed will be finished.
If the relation between GH and Klotho “has been perfectly described”, what was the rationale for repeating this?
Just to support the aspect of the redox balance in arteriogenesis and the possible effect of GH in this aspect. The key aspect of Klotho’s performance by modifying the redox balance may support the fact that GH does the same, as it is thought that Klotho may acts through GH in many tissues as arteries.
There are several unusual terms, e.g. “vein cava” should read “vena cava”.
It has been corrected now.
Round 2
Reviewer 1 Report
The authors responded to the suggestions. The current version is acceptable.
Author Response
Thank you for your comments.
We reduced the size of the text and references, we deleted Figure 7, and we added a new Table (Table 2 now).
Reviewer 2 Report
The authors have made some minor changes to the manuscript that failed to address previous concerns. This leaves this reviewer no option other than reiterate key concerns (for a detailed comments please refer to the previous comments).
The manuscript lacks molecular insights into mechanisms underlying arteriogenesis (as one would anticipate it from a review on the topic). The clinical data appear to be rather preliminary and reporting does not meet accepted standards. For instance, group A and B have not been defined, nor patient characteristics for each group are presented. The patient numbers in table 2 and figures do not match.
If the results from the GHAS trial will be presented in the future, as indicated by the authors, what could one possibly learn from fragmented data? Should not one report the findings upon completing the “molecular studies”, as explained in the rebuttal? In their rebuttal, the authors apparently agreed that “no conclusion can be drawn” from n=2-3. Then how would such data support any “fact”?
The text provide little help linking the listed molecules. For instance, if redox balance is critical (as suggested by the authors), how could this be linked to B cells? Changing colors in the figure does not address previous concerns. For instance, what does “hROS” “orange arrow induces transformation” refer to?
What is the rationale for discussing observations that were not the subject of the review?
Statistically non-significant changes should not be described as trends.
As opposed to the rebuttal, Table 1 still does not explain what do “0” and “2” refer to?
Author Response
The authors have made some minor changes to the manuscript that failed to address previous concerns. This leaves this reviewer no option other than reiterate key concerns (for a detailed comments please refer to the previous comments).
The manuscript lacks molecular insights into mechanisms underlying arteriogenesis (as one would anticipate it from a review on the topic). The clinical data appear to be rather preliminary and reporting does not meet accepted standards. For instance, group A and B have not been defined, nor patient characteristics for each group are presented. The patient numbers in table 2 and figures do not match.
First, the objective of this manuscript is not to do a review of arteriogenesis, but to highlight those aspects in which GH therapy could participate to support the role of this hormone in the former process. We believe that several molecular key aspects of arteriogenesis have been discussed as real eNOs role or the role of redox stress, trying to combine data from other authors and our own data based in a clinical trial, and thus, in a real scenario. As far as we know, the role of redox balance in vascular growth and vascular homeostasis appears poorly in scientific articles, both in review and intervention articles. To give for the first time data from patients with severe ischemia that support this role is something to be highly considered and new.
The main characteristics of the patients had been now added in Table 2 as required by referee two. Group A or B has been now defined and explained in the text and legends. Every aspect has been explained to not create confusion.
Thanks for noticing the mistake in Figure 6C. In the final samples of placebo patients (final B): instead of n=14, should appear n= 13), now corrected.
We also changed the position of the Tables, since there was another mistake in our previous submission after the reviewer's consideration: now table 1 and 2 are tables built with characteristics of patients from the GHAS trial and Table 3 is dedicated to biomarkers, adding the CRP. We really regret these mistakes.
The fact that the number of patients does not match in some results and Figures has a simple explanation. In clinical trials like this, some of the patients died before the trial is finished, suffer a limb amputation or sometimes refuse to perform the biopsy or the blood sample. This is the reason why in Tables and Figures always appear the sample size associated with the specific results.
If the results from the GHAS trial will be presented in the future, as indicated by the authors, what could one possibly learn from fragmented data? Should not one report the findings upon completing the “molecular studies”, as explained in the rebuttal? In their rebuttal, the authors apparently agreed that “no conclusion can be drawn” from n=2-3. Then how would such data support any “fact”?
Figure 7 has been deleted now to not create controversy. This figure was just an example to support the action of GH in EPCs stimulation in a real setting. However, given that this action of GH has been demonstrated yet in healthy subjects, and that the number of final results are few to draw any conclusion, we have decided to eliminate it.
The text provide little help linking the listed molecules. For instance, if redox balance is critical (as suggested by the authors), how could this be linked to B cells? Changing colors in the figure does not address previous concerns. For instance, what does “hROS” “orange arrow induces transformation” refer to?
This commentary mixes different things. First, in figure two orange arrow indicates that an increase in hROS induces the transformation of NO into NOOO- (now indicated in the legend of this Figure. Second, we think that when the reviewer talks about B cells, he is indicating B lymphocytes. If this is true, in the text is just described that B Lymphocytes are ones of the immune cells that express GH and its GHR, but it cannot be inferred that B cells play a role in arteriogenesis, as it was clearly exposed in the previous comments. We are not linking B lymphocytes to redox. Nevertheless, the reference to B cells has been eliminated to not create confusion.
What is the rationale for discussing observations that were not the subject of the review?
Statistically non-significant changes should not be described as trends.
As opposed to the rebuttal, Table 1 still does not explain what do “0” and “2” refer to?
Indeed, there is no rationale for discussing observation out of the scope of this review. Most of the commentaries related to these observations have been eliminated to make shorter the text. Some of the examples that appeared before just wanted to support some of the ideas expressed, taking into the account that the neovascularization that takes place in tumors has to be some similarities with that in ischemic conditions. Besides, some of the insights about arteriogenesis have been inferred from studies in tumors.
No reference to trends appears now in the text.
“0” and “2” in table 1 (now 3) about TNF-alpha have been changed by baseline and final results.
Thank you for your suggestions and excuse us, please, for the previous mistakes